# Loss of Katnal2 leads to ependymal ciliary hyperfunction and autism-related phenotypes in mice

Ryeonghwa Kang[1,2☯], Kyungdeok Kim[2☯], Yewon Jung[1☯], Sang-Han Choi[3,4], Chanhee Lee[3], Geun Ho Im[3], Miram Shin[5], Kwangmin Ryu[1], Subin Choi[5], Esther Yang[6], Wangyong Shin[2], Seungjoon Lee[2], Suho Lee[2], Zachary Papadopoulos[7], Ji Hoon Ahn[8], Gou Young Koh[8], Jonathan Kipnis[7,9,10], Hyojin Kang[11], Hyun Kim[6], Won-Ki Cho[1], Soochul Park[5], Seong-Gi Kim[3,4], Eunjoon Kim[1,2]*

1 Department of Biological Sciences, Korea Advanced Institute of Science and Technology (KAIST), Daejeon, Korea, 2 Center for Synaptic Brain Dysfunctions, Institute for Basic Science (IBS), Daejeon, Korea, 3 Center for Neuroscience Imaging Research, Institute for Basic Science (IBS), Suwon, Korea, 4 Department of Biomedical Engineering, Sungkyunkwan University, Suwon, Korea, 5 Department of Biological Sciences, Sookmyung Women's University, Seoul, Korea, 6 Department of Anatomy, Biomedical Sciences, College of Medicine, Korea University, Seoul, Korea, 7 Neuroscience Graduate Program, School of Medicine, Washington University in St. Louis, St. Louis, Missouri, United States of America, 8 Center for Vascular Research, Institute for Basic Science (IBS), Daejeon, Korea, 9 Brain Immunology and Glia (BIG) Center, Washington University in St. Louis, St. Louis, Missouri, United States of America, 10 Department of Pathology and Immunology, School of Medicine, Washington University in St. Louis, St. Louis, Missouri, United States of America, 11 Division of National Supercomputing, Korea Institute of Science and Technology Information (KISTI), Daejeon, Korea

☯ These authors contributed equally to this work.
* kime@kaist.ac.kr

**Data Availability Statement:** Raw RNA-Seq data are available as GSE219228 at Gene Expression Omnibus (GEO), National Center for Biotechnology Information (NCBI). The custom MATLAB code

## Abstract

Autism spectrum disorders (ASD) frequently accompany macrocephaly, which often involves hydrocephalic enlargement of brain ventricles. Katnal2 is a microtubule-regulatory protein strongly linked to ASD, but it remains unclear whether Katnal2 knockout (KO) in mice leads to microtubule- and ASD-related molecular, synaptic, brain, and behavioral phenotypes. We found that Katnal2-KO mice display ASD-like social communication deficits and age-dependent progressive ventricular enlargements. The latter involves increased length and beating frequency of motile cilia on ependymal cells lining ventricles. Katnal2-KO hippocampal neurons surrounded by enlarged lateral ventricles show progressive synaptic deficits that correlate with ASD-like transcriptomic changes involving synaptic gene down-regulation. Importantly, early postnatal Katnal2 re-expression prevents ciliary, ventricular, and behavioral phenotypes in Katnal2-KO adults, suggesting a causal relationship and a potential treatment. Therefore, Katnal2 negatively regulates ependymal ciliary function and its deletion in mice leads to ependymal ciliary hyperfunction and hydrocephalus accompanying ASD-related behavioral, synaptic, and transcriptomic changes.

used for the analysis is available in https://github.com/sjleen/Cilia-stroke-analysis.

**Funding:** This work was supported by the National Research Foundation of Korea grants RS-2023-00272290 (to HK), 2022M3A9B6082673 (to WKC), and NRF-2021R1A2C3011919 (to SP) funded by the Korean government and the Institute for Basic Science (IBS) (IBS-R015-D1 to SGK and IBS-R002-D1 to EK). The funders had no role in study design, data collection and analysis, decision to publish, or preparation of the manuscript.

**Competing interests:** The authors have declared that no competing interests exist.

**Abbreviations:** ASD, autism spectrum disorder; AUC, areas under the curve; BP, biological process; CBV, cerebral blood volume; CC, cellular component; CSF, cerebrospinal fluid; DEG, differentially expressed gene; EPSC, excitatory postsynaptic current; FDR, false discovery rate; fps, frames per second; GE-EPI, gradient-echo echo-planar image; GO, gene ontology; GSEA, gene set enrichment analysis; ICP, intracranial pressure; KO, knockout; LW, lateral wall; MF, molecular function; MRI, magnetic resonance imaging; MT, microtubule; NES, normalized enrichment score; PBS, phosphate-buffered saline; PCR, polymerase chain reaction; SEM, scanning electron microscopy; USV, ultrasonic vocalization; WT, wild type.

## Introduction

Autism spectrum disorders (ASD) represent a neurodevelopmental disorder characterized by social deficits and repetitive behaviors. Macrocephaly with increased brain size is frequently associated with ASD and neurodevelopmental disorders [1,2]. Macrocephaly in ASD can involve hydrocephalus [3,4], wherein ventricular cavities containing the cerebrospinal fluid (CSF) are abnormally enlarged [5,6]. Hydrocephalus can arise from various causes; among them are impairments in the circulation of CSF across ventricular structures, which is facilitated by the synchronous beating of the motile cilia belonging to ependymal cells lining ventricular walls [7–11]. Although primary and motile cilia and related genes have been implicated in hydrocephalus [9,10,12], it remains unknown whether ciliopathies could underlie ASD-related hydrocephalus and macrocephaly. Furthermore, previous studies have mostly focused on conditions in which the ependymal cilia are malfunctioning, despite the possibility that an increased ependymal ciliary function can also lead to hydrocephalus.

Microtubules (MTs) regulate various cellular morphologies and functions, including MT-dependent cellular motility and protein transport [13]. Katnal2 localizes to MT-based structures, such as primary and motile cilia; it is thought to regulate MTs, based on its structural similarity to the known MT-severing proteins, Katna1 and Katnal1, although direct MT-severing activity has not been observed [14–17]. Previous studies on in vivo functions of Katnal2 have mainly focused on how developmental processes are affected by Katnal2 deletion or knockdown. Katnal2 knockout (KO) in mice inhibits MT-related spermatogenic processes (i.e., spermiogenesis) [18,19]. Katnal2 knockdown in the developing Xenopus embryo suppresses epithelial cell ciliogenesis and telencephalic development [14]. Katnal2 KO in zebrafish leads to delayed embryonic development and abnormal social behaviors [20]. An acute Crispr-mediated Katnal2 KO in the neonatal mouse hippocampus suppresses the dendritic growth of dentate granule cells [21]. Clinically, Katnal2 has been strongly implicated in ASD [22–29]. However, it remains unclear whether and how Katnal2 KO in mice leads to ASD-related behavioral, brain, synaptic, and molecular/cellular phenotypes.

We found that mice lacking Katnal2 proteins, which are highly expressed in ependymal cells lining the brain ventricles, show social communication deficits and age-dependent progressive enlargements of brain ventricles (ventriculomegaly). These changes involved abnormal lengthening and excessive beating of motile cilia on ependymal cells lining ventricles. These changes accompanied impaired synaptic functions involving ASD-like transcriptomic changes and synaptic gene down-regulations. Importantly, early postnatal re-expression of Katnal2 prevented the ventricular and behavioral phenotypes in adult mutant mice. Therefore, Katnal2 negatively regulates ependymal ciliary development and function, and Katnal2 KO can lead to ASD-related behavioral, ventricular, synaptic, and transcriptomic changes. More generally, our study demonstrates that hydrocephalus can be caused by both limited and excessive ependymal ciliary functions.

## Results

### Abnormal social communication and mounting in Katnal2-KO mice

To investigate in vivo functions of Katnal2, which is a MT-related protein that has been strongly implicated in ASD (**S1A Fig**), we used a mouse embryonic cell line (EUCOMM, MAE-4330) to generate Katnal2-KO mice carrying a frame-shifting homozygous deletion of exon 3 (**S1B and S1C Fig**). Katnal2-KO mice lacked detectable Katnal2 protein in homozygotes and showed normal mendelian ratios and body weights (**S1D and S1E Fig**). Katnal2-KO brains were largely normal in cortical layer structures, as assessed using the markers NeuN

(neurons), Cux1 (layer 2/3), and Foxp2 (layer 6) (**S1F and S1G Fig**). Katnal2 KO did not affect dendritic morphology, which is strongly affected by microtubule structures, as determined by the Sholl analysis of CA1 hippocampal neurons in Katnal2-KO mice crossed with Thy1-EGFP mice [30] (**S1H Fig**). In addition, Katnal2 KO did not affect the length of the primary cilia, a microtubule-based structure protruding from the cell surface and is implicated in neuronal signaling [31], in cultured hippocampal neurons, as determined by immunostaining for Arl13b and NeuN, markers of primary cilia and neurons, respectively (**S1I Fig**).

Given that Katnal2 is associated with ASD, we first tested if Katnal2-KO mice show autistic-like behavioral deficits. Katnal2-KO mice showed normal social approach and social novelty recognition in the three-chamber test, as compared with wild-type (WT) mice (**S2A Fig and S1 Data** [statistical details and numerical data]). Katnal2-KO and WT mice did not display any genotype-related difference in the direct social interaction test (**S2B Fig**). Katnal2-KO mice also acted normally in unidirectional direct social interactions involving a stranger mouse of a different coat color (**S2C Fig**).

In tests measuring social communication through ultrasonic vocalizations (USVs), an adult male Katnal2-KO mouse emitted courtship USVs of increased frequency and duration upon encounter with a female stranger mouse (**S2D Fig**). To better understand this change, we analyzed direct social interactions occurring in the courtship arena. We found no genotype-related difference in uni- or bidirectional social interactions (male-to-female, female-to-male, and reciprocal) (**S2E Fig**). Intriguingly, however, male Katnal2-KO mice exhibited reduced mounting success (**S2F Fig**). Katnal2-KO mice displayed normal repetitive self-grooming (**S2G Fig**).

In other behavioral tests, Katnal2-KO mice showed normal levels of locomotor activity in the open-field test and normal levels of anxiety-like behaviors in the open-field, elevated-plus maze, and light-dark tests (**S3A–S3C Fig**). Katnal2-KO mice showed normal auditory function in the acoustic startle test and normal motor coordination in the rotarod test (**S3D and S3E Fig**). Lastly, Katnal2-KO mice performed normally in learning and memory tests, including the Morris water maze, novel object-recognition, and contextual-fear conditioning tests (**S3F–S3H Fig**). These results suggest that Katnal2-KO mice show selective abnormalities in social communication at adult stages.

## Progressive ventricular enlargements in Katnal2-KO mice

While performing the slice-staining experiments mentioned above, we noticed that the ventricles in Katnal2-KO brains were abnormally enlarged, suggestive of hydrocephalus. We thus quantitatively compared the brain areas and ventricular sizes of WT and Katnal2-KO brains at 3 different postnatal stages (P7, P28, and P70).

Pronounced ventricular enlargements in the two lateral ventricles of Katnal2-KO brains were observed at P28 and P70 (**Fig 1A–1C**). Some mutant mice (<10%) showed visually detectable macrocephaly with abnormal head shapes at approximately P28 (**Fig 1B**) and tended to die at approximately P30–40. Total brain areas in coronal slices were moderately increased at P70 but not at P28 in Katnal2-KO mice (**Fig 1A and 1C**). At P7, lateral ventricular enlargements were moderate compared with those at P28 and P70 (**S4A Fig**).

Magnetic resonance imaging (MRI) analysis revealed similar ventricular enlargements in the volumes of lateral ventricles among Katnal2-KO mice (3 months) (**Fig 1D**). In contrast, the intracranial volumes of mutant mice (encompassing all ventricular and non-ventricular brain regions) remained largely unchanged, with very moderate increases in selective brain regions (**Figs 1D and S4B**). These results collectively suggest that Katnal2 KO in mice leads to

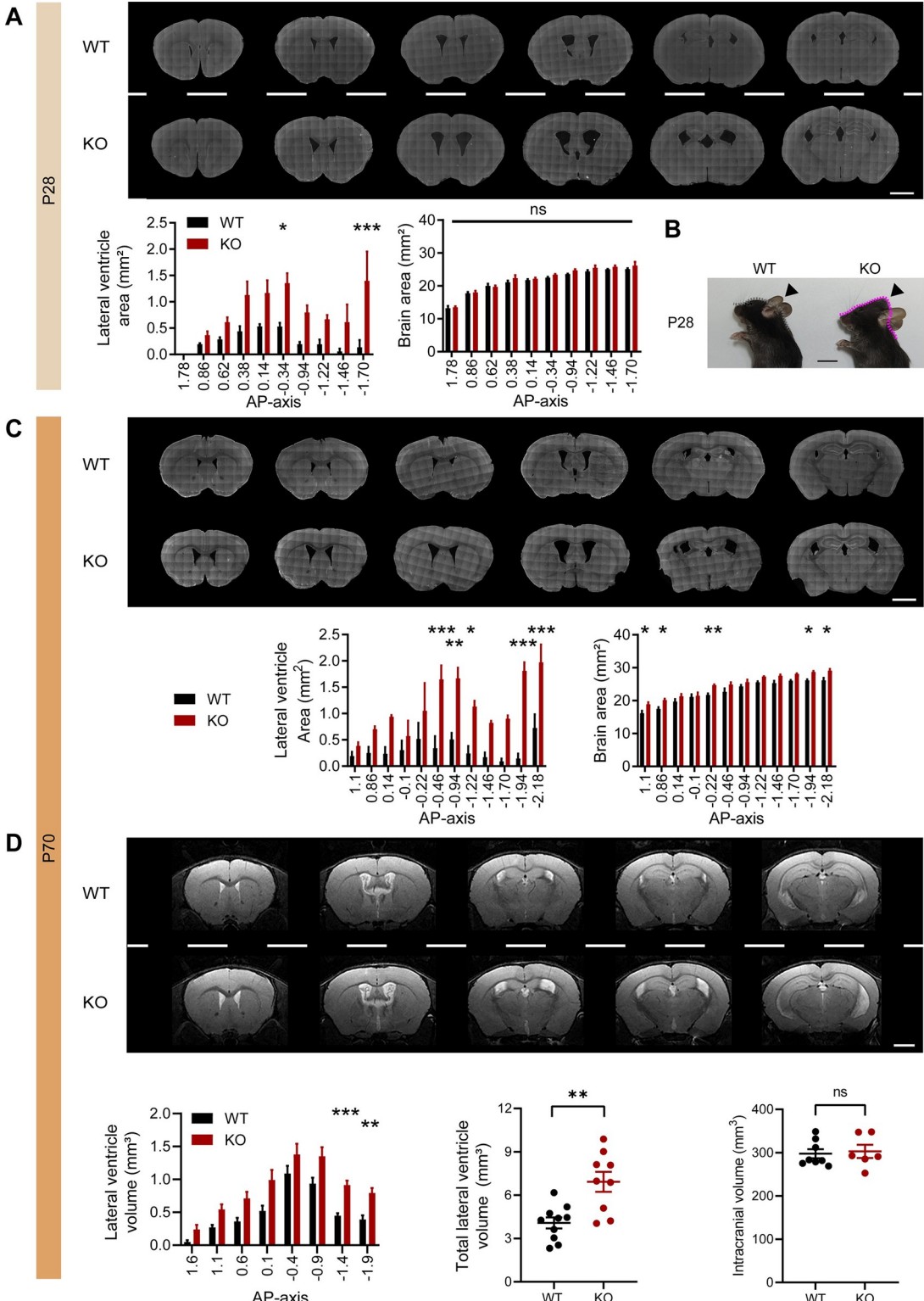

**Fig 1. Age-dependent progressive ventricular enlargements in Katnal2-KO mice.** (A–C) Increased areas of lateral ventricles in Katnal2-KO mice at P28 and P70, as shown by area measurements derived from coronal brain slices. Note that brain areas are moderately increased at P70 but not at P28. The image in (B) shows an example of the visually detectable abnormalities in the external regions of the head in some (<10%) mice (P28), likely reflecting severe hydrocephalus. AP axis, anterior-posterior axis. Scale bar, 2 mm. ($n$ = 3 mice [WT-P28], 3 [KO-P28]; 3 [WT-P70], 3 [KO-P70], two-way RM-ANOVA with Sidak's test). (D)

Increased volumes of lateral ventricles with unaltered intracranial volumes (total brain volume encompassing ventricular + non-ventricular brain regional volumes) in Katnal2-KO mice (3 months), as shown by MRI volumetric analyses. Scale bar, 2 mm. (*n* = 8 mice [WT-intracranial volume], 6 [KO-intracranial volume], Student's *t* test; *n* = 10 mice [WT-lateral ventricle volume], 9 [KO-lateral ventricle volume], two-way RM-ANOVA with Sidak's test). Data values represent means ± SEM. Significance is indicated as * (<0.05), ** (<0.01), *** (<0.001), or ns (not significant). Statistical results and numerical data values can be found in **S1 Data**. KO, knockout; MRI, magnetic resonance imaging; WT, wild type.

age-dependent and progressive ventricular enlargements with moderate effects on total brain areas and that severe macrocephaly is observed in some cases.

## Katnal2 localization in ventricular ependymal cells

The hydrocephalus observed in Katnal2-KO mice may involve impaired circulation of the CSF through the ventricular system. We thus tested if Katnal2 proteins are distributed to ventricle-related structures of the mouse brain using X-gal staining of brain slices from Katnal2-KO mice expressing Katnal2-β-galactosidase fusion proteins (**S1B Fig**).

Katnal2 signals at P21 were detected in multiple brain regions, including ventricle-related regions, such as the linings of the lateral and third ventricles where ependymal cells are located (**Figs 2A** and **S5A**). Katnal2 was also detected in the choroid plexus, a CSF-producing vascular structure in ventricles, as well as in other brain regions, such as the septum, hippocampus, amygdala, and hypothalamus. At P56, Katnal2 was additionally detected in cortical layers (**S5B Fig**).

In a combined fluorescence in situ hybridization and immunofluorescence staining, Katnal2 mRNAs colocalized with S100 (ependymal cell marker) and acetylated β-tubulin (cilia marker) in the ependymal cells lining lateral/third ventricles (**Fig 2B and 2D**). Katnal2 mRNAs also colocalized with FoxJ1 (choroid plexus marker) in the choroid plexus of lateral/third ventricles (**Fig 2C and 2E**). These results indicate that Katnal2 is expressed in the choroid plexus and ependymal cells, which regulate CSF production and CSF circulation, respectively, in the ventricular system [7,32].

## Lengthened ependymal cilia and increased ciliary beating function and frequency in the Katnal2-KO brain

Katnal2 is an MT-related protein. Notably, ependymal cilia comprise an MT-based 9+2 axonemal structure, and hydrocephalus can be caused by ciliary impairments [9,33]. We thus tested whether Katnal2 KO leads to altered ependymal ciliary structures in Katnal2-KO mice using scanning electron microscopy (SEM).

SEM analysis indicated that the ciliary length was abnormally increased in Katnal2-KO mice (**Figs 3A, 3B, and S6**), suggesting that Katnal2 negatively regulates the length of the axonemal structure in ependymal cilia.

To determine functional abnormalities in Katnal2-KO ependymal cilia, we first measured the flow rate of fluorescent beads moving across lateral ventricular walls in a whole-mount en face brain preparation (**Fig 3C**). Time-lapse imaging indicated that there are significant increases in the flow rates in the three tested flow pathways (**Fig 3D–3G**). In a control experiment, the bead flow rate was substantially decreased in the presence of pneumolysin (**Fig 3H**), a pore-forming pneumococcal toxin known to suppress ependymal ciliary beating and CSF flow in an irreversible manner [34–37]. These results suggested that Katnal2 KO enhanced ependymal ciliary function.

We next measured ependymal ciliary beating frequency in brain slices using a high-speed (approximately 1,000 frames/sec) camera and video clip analysis (**Fig 3I and S1 Movie**).

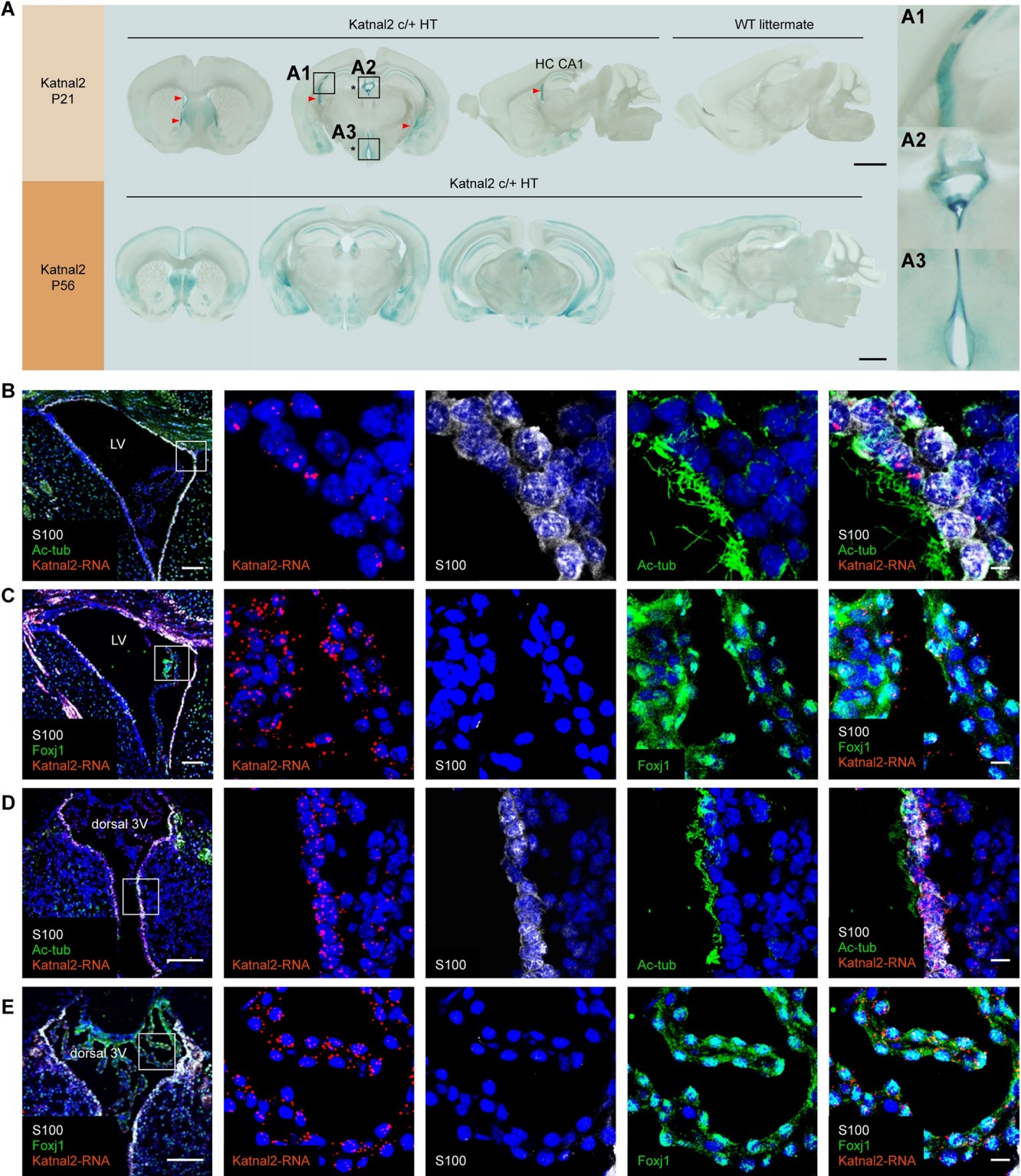

**Fig 2. Katnal2 localization in ventricular ependymal cells.** (A) Distribution patterns of Katnal2 proteins in ventricle-related structures (ventricles [arrowheads] and choroid plexus [the bottom structure in enlarged inset a2]) in the mouse brain, as revealed by X-gal staining of Katnal2-β-galactosidase fusion proteins expressed in Katnal2-KO mice with the β-geo cassette left intact (P21 and P56). Note that Katnal2 is additionally detected in cortical layers at P56 but not at P21. Enlarged insets: a1, lateral ventricle; a2, dorsal third ventricle; a3, ventral third ventricle. See also **S5 Fig** for additional images. Scale bar, 1 mm. (B–E) Localizations of Katnal2 mRNAs in the ependymal cells and choroid plexus in the lateral ventricles (b, c; LV) and dorsal third ventricles (d, e;

D3V) of WT mice (P56), as shown by combined fluorescence in situ hybridization for Katnal2 mRNAs and immunofluorescence staining for S100 (ependymal cell marker), acetylated β-tubulin (cilia marker; Ac-tub), and FoxJ1 (choroid plexus marker). Scale bar, 100 μm (left) and 10 μm (magnified images, right). KO, knockout; WT, wild type.

Intriguingly, Katnal2-KO cilia showed an increased ciliary beating frequency in lateral ventricles (**Fig 3I**). This increase in beating frequency, together with the lengthened cilia, suggests that Katnal2 KO may enhance ependymal ciliary function, leading to a stronger propulsion of CSF (**Fig 3J**).

Increased CSF propulsion would exert greater hydraulic turbulances on downstream CSF channels. Indeed, we found that the flow rates of CSF in the cerebral aqueduct, which links the third and fourth ventricles, was increased in the brains of anesthetized Katnal2-KO mice, as measured by MRI (**Fig 3K**). However, the intracranial pressure was not changed in the brains of Katnal2-KO mice (**Fig 3L**), suggestive of a counteraction between increased hydraulic turbulence and increased ventricular volume reached at an equilibrium. We performed these experiments mainly using Katnal2-KO mice at ages >P28 because ependymal ciliary maturation is established at around P21–28 [38–40].

These results collectively suggest that Katnal2 KO enlengthens ependymal ciliary length and enhances ciliary beating function and frequency, leading to increased CSF propelling down the stream.

## Progressive synaptic deficits in the Katnal2-KO hippocampus

Hydrocephalus can compress brain parenchyma, including perivascular spaces, and cause pathophysiological changes in compressed neurons and axons [41,42], as exemplified by impaired synaptic plasticity in animal models of chronic hydrocephalus [43]. We thus measured cerebral blood volumes (CBVs) in the Katnal2-KO brain (**Fig 4A–4D**) and synaptic transmission and plasticity (**Fig 4E–4N**) in the Katnal2-KO hippocampus, a brain region surrounded by the enlarged lateral ventricles.

CBV-weighted signals, measured by $T_2^*$-weighted single-shot gradient-echo echo-planar images (GE-EPI) combined with acute hypoxic nitrogen stimulus [44], were decreased in various brain regions, including the hippocampus (**Figs 4A–4D and S7A**). In contrast, CBV-weighted signals around the choroid plexus were unaffected (**S7B and S7C Fig**), suggesting that CSF production, relying on the blood flow to the choroid plexus [45], was not changed. In addition, immunoblot analysis of choroid plexus lysates indicated the lack of genotype difference in the levels of proteins known to regulate CSF secretion such as ion co-transporters ($Na^+$-$K^+$-ATPase and anion exchanger 2/AE2; marking apical and basolateral epithelial membranes, respectively) and the water channel aquaporin-1 (AQP1) [46,47] (**S7D Fig**) further suggesting that CSF production is not changed in Katnal2-KO mice. There were no changes in the drainage of CSF from the brain to the lymph nodes outside the brain, as measured by the signals of fluorescent ovalbumin injected into lateral ventricles and retrieved in the deep cervical lymph nodes (**S7E and S7F Fig**).

Measurements of hippocampal synaptic functions in juvenile (P20–33) Katnal2-KO mice indicated normal long-term potentiation induced by theta-burst stimulation (TBS-LTP) at Schaffer collateral-CA1 (SC-CA1) synapses (**Fig 4E**). In addition, there was no change in basal synaptic transmission (AMPA receptor-mediated excitatory postsynaptic currents [EPSCs]), presynaptic release (paired-pulse facilitation), or the NMDA/AMPA ratio (ratio of NMDA receptor- and AMPA receptor-mediated EPSCs) (**Fig 4F–4H**). Neuronal excitability in CA1 neurons was also normal, as shown by the input-firing curve (**Fig 4I**).

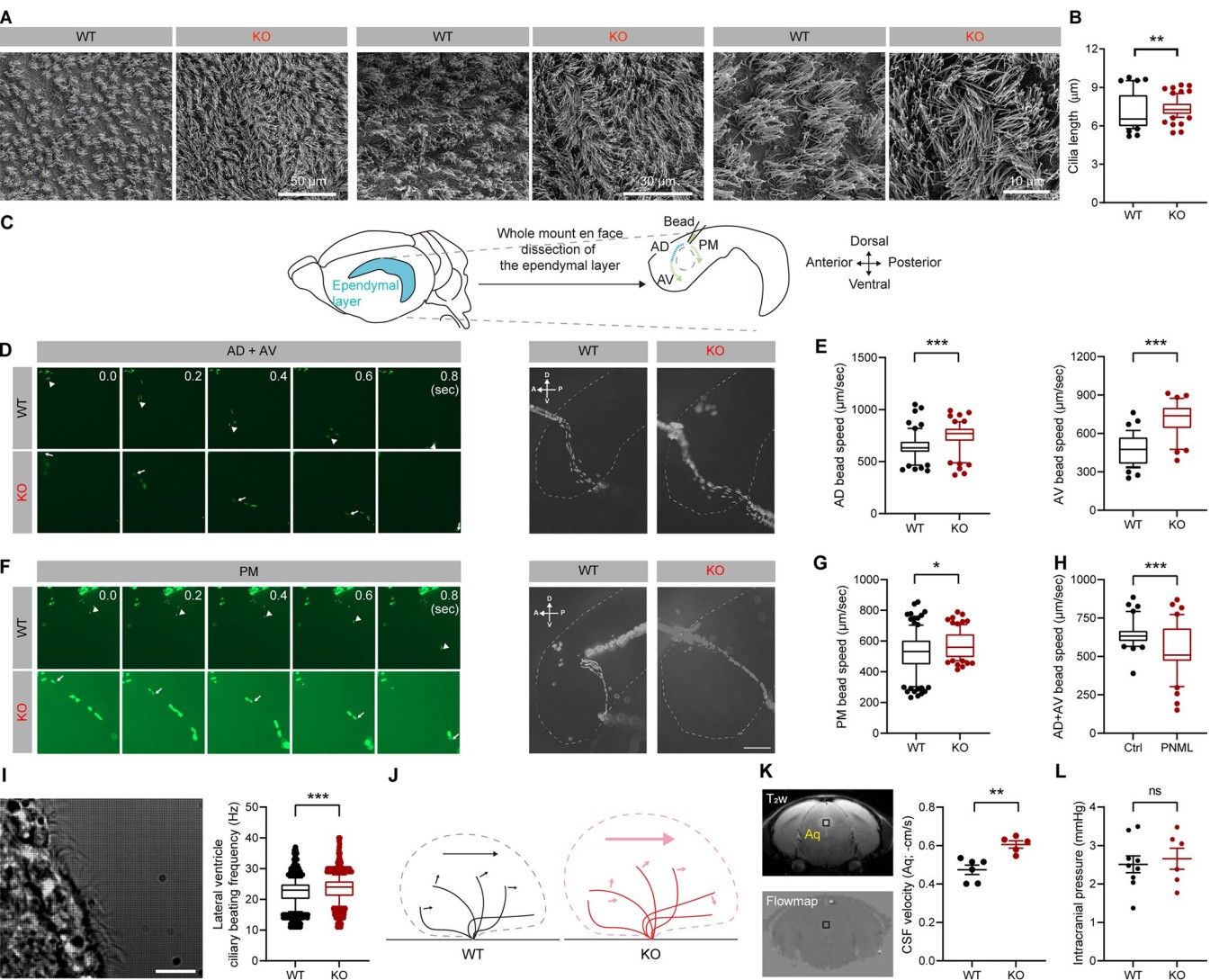

**Fig 3. Lengthened ependymal cilia and increased ciliary beating function and frequency in the Katnal2-KO brain.** (A and B) Increased length of ependymal motile cilia in Katnal2-KO mice (P28–33), as determined from SEM images of lateral ventricular walls. (*n* = 42 cilia from 6 images from 6 mice [WT], 77, 11, 11 [KO], Mann–Whitney test). (C–H) Enhanced ependymal ciliary function in Katnal2-KO mice (P31–40), as shown by bead flow assays using the ependymal tissue of lateral ventricular walls. (C) Schematic diagram showing the movement of fluorescent beads on live lateral ventricular walls. The bead flow follows the anterior-dorsal (AD), anterior-ventral (AC), or posterior-medial (PM) direction around the adhesion area. (D and F) High-speed video imaging analysis of each fluorescent bead at different time points. The movement of each bead was marked by arrowheads. Five consecutive frames taken by 200 msec intervals were merged into a single image. Scale bar, 200 μm. (E and G) Quantification of the speed of microbeads. (*n* = 60 beads from 4 mice [WT-AD], 39, 4 [WT-AV], 62, 4 [KO-AD], 33, 4 [KO-AV], 116, 4 [WT-PM], and 82, 4 [KO-PM], Student's *t*-test [AV bead speed], Mann-Whitney test [AD bead speed, PM bead speed]). (H) A decreased speed of microbeads in AD + AV regions after a 10-min pretreatment of brain slices with pneumolysin (PNML; 0.5 μg/ml) (*n* = 40 beads from 3 mice for pneumolysin and control [without pneumolysin] groups, Mann–Whitney test). (I) Increased ciliary beating frequency in lateral ventricles of Katnal2-KO mice (P28–42), as measured by high-speed (approximately 1,000 frames/sec) time-lapse imaging and quantification of the beating frequency. Scale bar, 10 μm. (*n* = 1,442 ROIs from 67 cilia videos from 10 mice [WT-lateral], 1,346, 72, 9 [KO-lateral], Permutation test). (J) A working hypothesis suggesting that increased ependymal ciliary length and beating frequency would enhance CSF propulsion. The big arrow indicates the overall propulsion power of the cilia, and the small arrows indicate the forward directions of each ciliary movement. The additional cilium in the Katnal2-KO mice indicates increased beating frequency. (K) Increased CSF flow rates in the cerebral aqueduct of Katnal2-KO mice (P28–42), as measured by MRI. (*n* = 6 mice [WT], 5 [KO], Student's *t* test). (L) Comparable ICPs in WT and Katnal2-KO mice (3 months). (*n* = 9 mice [WT], 6 [KO], Student's *t* test). Data values represent means ± SEM. Significance is indicated as * (<0.05), ** (<0.01), *** (<0.001), or ns (not significant). Statistical results and numerical data values can be found in **S1 Data**. CSF, cerebrospinal fluid; ICP, intracranial pressure; KO, knockout; MRI, magnetic resonance imaging; WT, wild type.

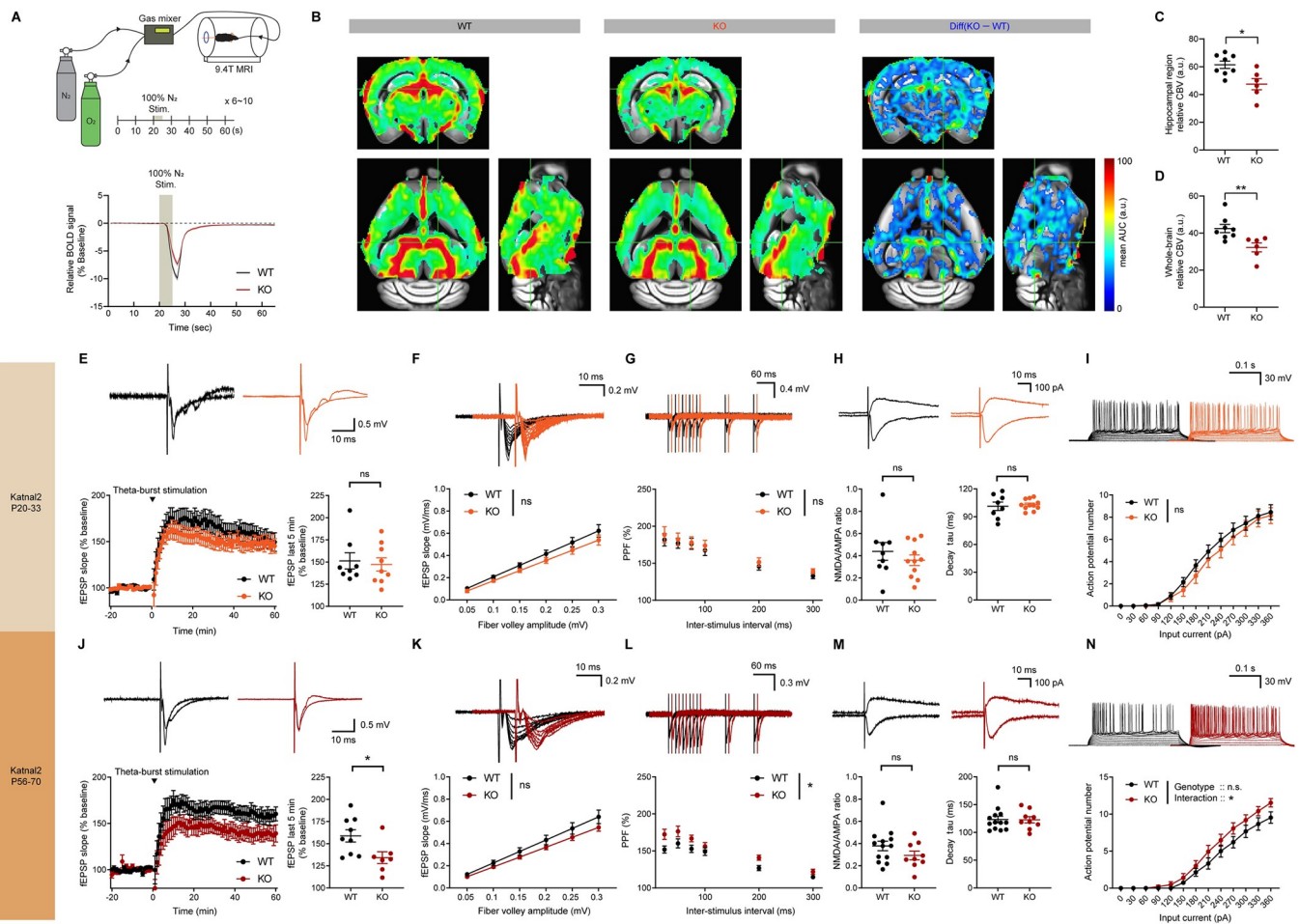

**Fig 4. Progressive synaptic deficits in the Katnal2-KO hippocampus.** (A–D) Decreased CBVs in various brain regions of Katnal2-KO mice, including the hippocampus surrounded by lateral ventricles, as compared with WT mice (3 months). CBV-weighted MRI measurements were performed with nitrogen stimuli in WT and Katnal2-KO mouse brains, and the relative CBV index was computed by the integration of the area under the curve (A). Note that the color-coded decreases in CBVs in Katnal2-KO mice occur in various brain regions (B, left and middle), although it seems to be greater in the mutant hippocampus (B, right). The results of the quantification of relative CBV indices are shown in (C) and (D) for the hippocampus and whole brain and in **S7A Fig** for various cortical and subcortical regions. ($n$ = 8 mice [WT], 6 [KO], Student's $t$ test). (E) Normal TBS-LTP at Katnal2-KO hippocampal SC-CA1 synapses (P26–33; last 5 min). ($n$ = 7 slices from 3 mice [WT], 9, 3 [KO], Student's $t$ test [last 5 min]). (F) Normal basal transmission at Katnal2-KO SC-CA1 synapses (P25-30), as measured by input-output curves. ($n$ = 7, 3 [WT], 8, 4 [KO], two-way RM-ANOVA). (G) Normal paired pulse facilitation at Katnal2-KO SC-CA1 synapses (P25-30). ($n$ = 9, 4 [WT], 10, 4 [KO], two-way RM-ANOVA). (H) Normal ratios of NMDAR- and AMPAR-mediated EPSCs (NMDA/AMPA ratios) at Katnal2-KO SC-CA1 synapses (P20-21). ($n$ = 9 neurons from 4 mice [WT], 11, 4 [KO], Student's $t$ test). (I) Normal neuronal excitability in Katnal2-KO CA1 pyramidal neurons (P26–30), as measured by input-firing curves. ($n$ = 20, 3 [WT], 14, 3 [KO], two-way RM-ANOVA). (J) Suppressed TBS-LTP at Katnal2-KO SC-CA1 synapses (P56–70). ($n$ = 9, 4 [WT], 7, 3 [KO], Student's $t$ test). (K) Normal input-output ratio at Katnal2-KO SC-CA1 synapses (P56–70). ($n$ = 9, 3 [WT], 9, 4 [KO], two-way RM-ANOVA). (L) Suppressed paired pulse facilitation at Katnal2-KO SC-CA1 synapses (P56–70). ($n$ = 10, 4 [WT], 10, 4 [KO], two-way RM-ANOVA with Sidak's test). (M) Normal NMDA/AMPA ratios at Katnal2-KO SC-CA1 synapses (P56–70). ($n$ = 13, 5 [WT], 9, 5 [KO], Mann–Whitney test [NMDA/AMPA ratio], Student's $t$ test [decay tau]). (N) Moderately increased excitability of Katnal2-KO CA1 pyramidal neurons (P56–70). ($n$ = 15, 4 [WT], 15, 4 [KO], two-way RM-ANOVA with Sidak's test). Data values represent means ± SEM. Significance is indicated as * ($<$0.05), ** ($<$0.01), *** ($<$0.001), or ns (not significant). Statistical results and numerical data values can be found in **S1 Data**. CBV, cerebral blood volume; EPSC, excitatory postsynaptic current; KO, knockout; MRI, magnetic resonance imaging; WT, wild type.

At adult stages (P56–70), TBS-LTP was suppressed at Katnal2-KO SC-CA1 synapses; there was no change in basal synaptic transmission or the NMDA/AMPA ratio, although presynaptic release and neuronal excitability were moderately increased (**Fig 4J–4N**). This suggests that the decrease in TBS-LTP at adult Katnal2-KO synapses does not involve a decrease in NMDAR function. These results collectively suggest that Katnal2 KO in mice suppresses synaptic functions in the adult but not juvenile hippocampus.

## Progressive synaptic and ASD-related transcriptomic changes in Katnal2-KO mice

To gain insights into the mechanisms underlying the behavioral, ciliary, and synaptic deficits observed in Katnal2-KO mice in an unbiased manner, we analyzed transcriptomic changes in Katnal2-KO brains at P21 and P70 by RNA-Seq analyses.

The RNA-Seq analyses revealed that there were small sets of differentially expressed genes (DEGs; $p < 0.05$ and fold-change $>1.5$) at both P21 and P70, as shown by volcano plots (28 up- and 11 down-regulated at P21, and 81 up- and 66 down-regulated at P70) (**S8 Fig** and **S2 Data** and **S3 Data**). DAVID analyses of these DEGs yielded no significant gene ontology (GO) terms, likely because there were relatively few DEGs.

We next attempted a gene set enrichment analysis (GSEA), which uses the whole list of ranked transcripts (i.e., by $p$ values) rather than a small fraction of transcripts above arbitrary cutoffs. This enables an unbiased exploration of altered biological functions [48]. Our results identified distinct biological functions as being altered in the Katnal2-KO transcriptomes at P21 and P70 (termed P21-Katnal2/WT and P70-Katnal2/WT transcripts, respectively). Specifically, P21-Katnal2/WT transcripts were positively, although moderately, enriched for vacuole- and extracellular matrix-related gene sets (meaning these genes were up-regulated), and negatively and moderately enriched for splicing- and cell morphogenesis-related gene sets in the cellular component (CC), biological process (BP), and molecular function (MF) domains, as shown by clusters of enriched gene sets generated using Cytoscape EnrichmentMap App [49] (**Fig 5A** and **S4 Data**).

At P70, Katnal2/WT transcripts were positively and strongly enriched for ribosome/mitochondria-related gene sets, and negatively and strongly enriched for synapse-related gene sets in the C5-CC, C5-BP, and C5-MF domains, as shown by top-five gene set lists and EnrichmentMap gene-set clusters [49] (**Fig 5B** and **S4 Data**). These results suggest that P21-Katnal2/WT and P70-Katnal2/WT transcripts are associated with distinct biological functions and that P70-Katnal2/WT transcripts display more strongly up-regulated ribosome/mitochondria-related genes and down-regulated synapse-related genes, as compared with P21-Katnal2/WT transcripts.

To determine if these transcriptomic changes are associated with ASD-related gene expression patterns, we performed GSEA using gene sets that are up- or down-regulated in ASD [50–54]. The results revealed that P21-Katnal2/WT transcripts were positively enriched for ASD-related gene sets that are up-regulated in ASD (DEG_Up and Co-Exp_Up), but not significantly enriched for those down-regulated in ASD (DEG_Down and Co-Exp_Down) (**Fig 5C** and **S5 Data**). In addition, P21-Katnal2/WT transcripts were negatively enriched for 2 of the 6 ASD-risk gene sets that are usually down-regulated in ASD (SFARI [all] and ASD_AutismKB, but not FMRP target, DeNovoMiss or DeNovoVariants) (**Fig 5C**). These results indicate that P21-Katnal2/WT transcripts are moderately enriched for ASD-related/risk gene sets (partial "ASD-like" pattern).

P70-Katnal2/WT transcripts showed a much stronger ASD-like pattern: They were positively enriched for Co-Exp_Up, negatively enriched for Co-Exp_Down, and negatively enriched for 5 out of the 6 ASD-risk gene sets (SFARI [all and high-confidence], FMRP target, DeNovoMiss, and DeNovoVariants, but not ASD_AutismKB) (**Fig 5C**). These results suggest that there are stronger ASD-like transcriptomic patterns in P70 Katnal2-KO mice relative to P21 Katnal2-KO mice.

When GSEA was performed against cell type-specific gene sets that are distinctly enriched in ASD [50–55], P21-Katnal2/WT transcripts were largely negatively enriched for neuron (both glutamate and GABA)-related gene sets and positively enriched for glia (astrocyte and

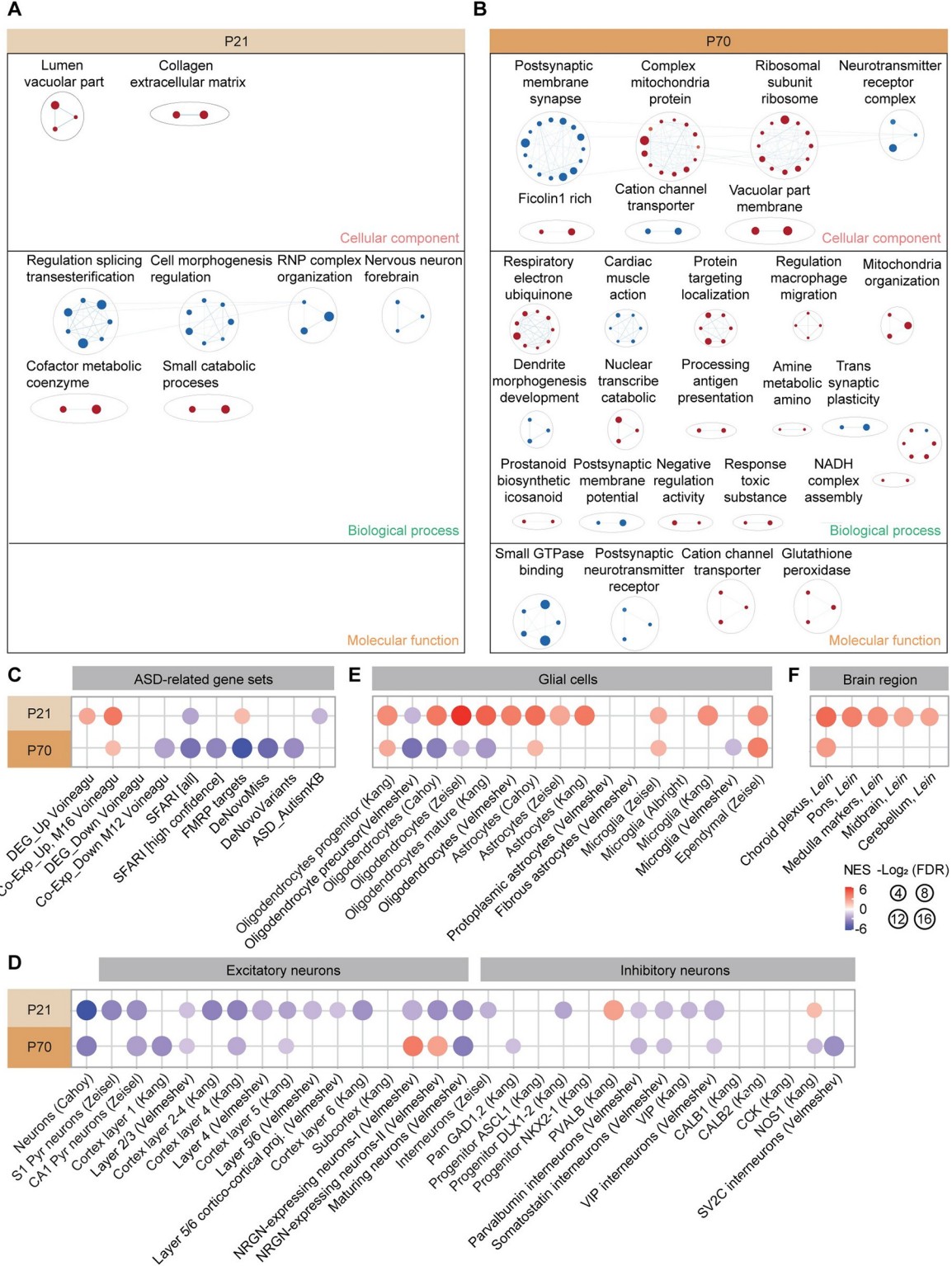

**Fig 5. Progressive synaptic and ASD-related transcriptomic changes in Katnal2-KO mice.** (A and B) GSEA results revealing the enrichment of P21- and P70-Katnal2/WT transcripts (whole brains) for biological functions, as shown by the list of top-five positively (red) and negatively (blue) enriched gene sets in the CC domain (left) and the clusters of enriched gene sets visualized using EnrichmentMap Cytoscape App (right). (C) GSEA results assessing Katnal2/WT transcripts for ASD-related gene sets that are up-regulated in ASD (DEG_Up and Co-Exp_Up), down-regulated in ASD (DEG_Down and Co-Exp_Down), and ASD-risk gene sets that

are usually down-regulated in ASD (SFARI [all], SFARI [high-confidence], FMRP target, DeNovoMiss, and DeNovoVariants, but not ASD_AutismKB). ($n$ = 5 mice for WT and KO, FDR < 0.05). (D) GSEA results assessing Katnal2/WT transcripts for neuron type-specific gene sets (excitatory and inhibitory neuronal) that are usually down-regulated in ASD. ($n$ = 5 mice for WT and KO, FDR < 0.05). (E) GSEA results assessing Katnal2/WT transcripts for glial cell type-specific gene sets that are usually down-regulated in oligodendrocytes and up-regulated in astrocytes and microglia in ASD. ($n$ = 5 mice for WT and KO, FDR < 0.05). (F) GSEA results assessing Katnal2/WT transcripts for brain region-related gene sets. ($n$ = 5 mice for WT and KO, FDR < 0.05). Raw RNA-Seq data and detailed GSEA results can be found in **S2 Data** and **S4 Data**. ASD, autism spectrum disorder; CC, cellular component; DEG, differentially expressed gene; FDR, false discovery rate; GSEA, gene set enrichment analysis; KO, knockout; WT, wild type.

microglia)-related gene sets; the directions were mainly in line with the those observed in ASD (ASD-like), although oligodendrocyte-related gene sets were positively enriched ("reverse-ASD" direction) (**Fig 5D and 5E**). Notably, P70-Katnal2/WT transcripts showed largely similar but weakened negative enrichments for neuron-related gene sets and positive enrichments for glia (including oligodendrocyte)-related gene sets (**Fig 5D and 5E**). Intriguingly, both P21 and P70 Katnal2/WT transcripts were positively enriched for an ependymal cell-related gene set (**Fig 5E**) and for a choroid plexus-related gene set (**Fig 5F**), which is in line with the strong expression of Katnal2 in ventricular structures (**Fig 2A**).

Therefore, the P21 and P70 transcriptomes in Katnal2-KO mice seem to display the following age-dependent and progressive changes: (1) weak changes in biological functions at P21, but strong biological changes at P70 (i.e., synaptic down-regulation); (2) stronger ASD-like expressional changes in ASD-related/risk gene sets at P70 compared with P21; (3) stronger ASD-like changes in neuronal and glial (astrocyte/microglia) gene expressions at P21 relative to P70, with the exception of oligodendrocytes (being more ASD-like at P70); and (4) ependymal and choroid plexus-related gene up-regulations at both P21 and P70.

## Katnal2 gene re-expression prevents ependymal ciliary, ventricular, and behavioral phenotypes

We hypothesized that if Katnal2 KO leads to the ventricular enlargements and behavioral deficits observed in Katnal2-KO mice, Katnal2 gene re-expression may rescue these deficits if attempted as early as possible. To assess this possibility, we employed AAV (PHP.eB)-mediated Katnal2 gene re-expression in the mutant mouse brain using early postnatal (P8–12) intracerebroventricular injection, at a time point when the ventricular enlargements were not yet evident. We then performed brain/ciliary morphological and behavioral analysis in adult mice (P56–70) (**Fig 6A**).

Early postnatal Katnal2 re-expression with stronger gene expression in ventricular regions (**Fig 6B**) restored ependymal ciliary length and ventricular enlargement in adult Katnal2-KO mice, as compared with controls (Katnal2-KO mice without Katnal2 re-expression and WT mice with EGFP/control overexpression) (**Fig 6C and 6D**). Early Katnal2 re-expression also partially prevented the abnormal increase in courtship USVs in adult Katnal2-KO mice, as supported by the comparable USV levels in Katnal2-re-expressing Katnal2-KO mice and WT mice with EGFP overexpression, although USV levels in Katnal2-KO mice with re-expression did not significantly differ from those in Katnal2-KO mice without re-expression (**Fig 6E**).

These results suggest that early postnatal Katnal2 re-expression in Katnal2-KO mice prevents ependymal ciliary lengthening, ventricular enlargements, and courtship USV deficits in adult mutant mice.

## Discussion

In the present study, we found that Katnal2 KO in mice leads to autistic-like abnormal social communications. These changes accompanied age-dependent progressive ventricular

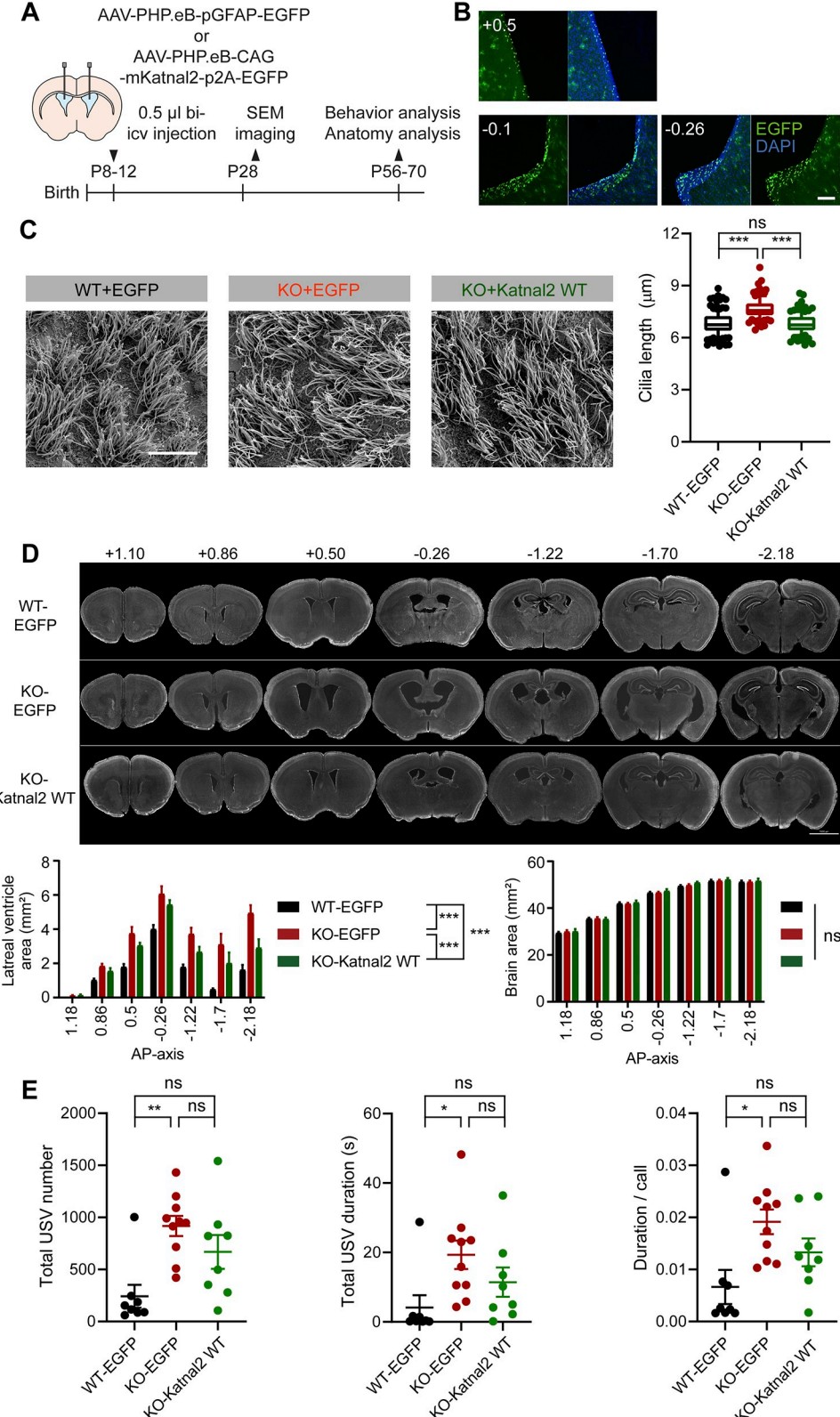

**Fig 6. Early Katnal2 gene re-expression prevents ciliary, ventricular, and behavioral phenotypes in adult Katnal2-KO mice.** (A) Schematic diagram of Katnal2 re-expression experiments in which PHP.eB.AAV-

pCAG-Katnal2-p2A-EGFP, or control virus (PHP.eB.AAV-pGFAP-EGFP; pGFAP for ependymal cell expression), was bilaterally injected into the lateral ventricles (intracerebroventricular/icv injection) of WT or Katnal2-KO mice at approximately P8–12, and the effects on ventricles and behaviors were determined at adult stages (approximately P56–70). (B) Examples of Katnal2 re-expression (marked by EGFP) in the lateral ventricles of Katnal2-KO brain (P56). The numbers above brain images indicate anterior-posterior positions. Scale bar, 200 µm. (C) Prevention of ependymal ciliary lengthening in the Katnal2-KO brain (P28–32) by early postnatal Katnal2 re-expression (P8–12), as compared with control conditions (Katnal2-KO-EGFP alone and WT-EGFP alone). ($n$ = 160 cilia from 16 images from 4 mice [WT-EGFP alone/control], 120, 12, 3 [KO-EGFP alone], and 120, 12, 3 [KO-rescue/Katnal2 re-expression], one-way ANOVA with Sidak's test). Scale bar, 10 µm. (D) Prevention of ventricular enlargement in the Katnal2-KO brain (P56–70) by early postnatal Katnal2 re-expression (P8–12), as compared with control conditions (Katnal2-KO-EGFP alone and WT-EGFP alone). Note that the brain areas are minimally affected by the treatments. ($n$ = 22 mice [WT-EGFP alone/control], 8 [KO-rescue/Katnal2 re-expression], and 20 [KO-EGFP alone], two-way RM-ANOVA with Sidak's test). Scale bar, 2 mm. (E) Partial prevention of excessive courtship USVs in Katnal2-KO mice (P56–70) by early postnatal Katnal2 re-expression (P8–12), as compared with control conditions (Katnal2-KO-EGFP alone [not significant] and WT-EGFP alone [not significant, meaning rescued]). ($n$ = 8 [WT-EGFP alone], 10 [KO-EGFP alone], and 8 [KO-rescue], one-way ANOVA with Sidak's test). Data values represent means ± SEM. Significance is indicated as * ($<0.05$), ** ($<0.01$), *** ($<0.001$), or ns (not significant). Statistical results and numerical data values can be found in **S1 Data**. KO, knockout; USV, ultrasonic vocalization; WT, wild type.

enlargements involving lengthened ependymal cilia and increased ciliary beating frequency. Similar progressive impairments were observed in synaptic functions and transcriptomic patterns. In addition, early postnatal Katnal2 re-expression prevented ependymal ciliary, ventricular, and behavioral deficits in mutant adults, causally linking Katnal2 KO with the phenotypes.

Katnal2-KO mice show social deficits, including abnormally increased USVs during courtship and reduced success of mounting after the initial social interactions in male mice, and avoidance of a second social interaction with a mutant male after the initial interaction in female mice. Because USV levels are correlated with social interaction levels in rodents [56], the excessive USVs in Katnal2-KO males in the face of decreased mounting success, which would substantially decrease reproductive success, may represent increased effort of male mutant mice for additional social interactions.

An important Katnal2-KO phenotype is age-dependent and progressive ventricular enlargement (ventriculomegaly), as shown by slice-staining and MRI results. The increased flow rate of CSF in the mutant cerebral aqueduct, measured by MRI, may reflect increased pressure of the CSF in the upstream lateral and third ventricles. However, the intracranial pressure was comparable in WT and Katnal2-KO mice, suggesting that an increase in CSF pressure might have been releaved by an increase in ventricular volume. Indeed, normal pressure is common in hydrocephalus, including the idiopathic normal-pressure hydrocephalus, which involves dilated ventricles, cognitive deficit, and neural symptoms [57–59]. Notably, the ventricular enlargement in the Katnal2-KO brain did not increase the total brain volume, likely because of the progressive nature of the ventriculomegaly. However, prominent macrocephaly and abnormal head shapes were observed in approximately 10% of Katnal2-KO mice, and these mice failed to survive beyond approximately P40. This suggests that Katnal2 KO leads to strong macrocephaly in some penetrant cases.

Ventricular enlargements in Katnal2-KO mice could be caused by impaired formation or function of ependymal cilia, considering that Katnal2 is highly expressed in ependymal cells and the choroid plexus in the mouse brain. Indeed, many of previous studies have reported impaired formation and function of ependymal cilia as the cause of hydrocephalus [60]. However, our data indicate that Katnal2 KO induces lengthened ependymal ciliary and enhanced ependymal ciliary function (increases in ciliary bead flow and beating frequency). These results suggest that Katnal2 negatively regulates ependymal ciliary length. Moreover, the lengthened ependymal cilia with increased beating frequency may strongly propel CSF down the stream and cause ventriculomegaly by exerting increased hydraulic stress on periventricular tissues,

although the measured intracranial pressure was normal, again suggestive of a pressure releave by increased ventricular volume. This proposed mechanism of ventriculomegaly differs radically from the previously reported mechanisms for hydrocephalus, such as impaired formation and function of ependymal cilia.

An important question would be whether the ventriculomegaly in Katnal2-KO mice has any functional influence on nearby brain regions, such as the hippocampus, which is surrounded by the enlarged lateral ventricles. Our MRI data indicate that the blood volumes in various mutant brain regions, including the hippocampus, are decreased. This could limit the supply of nutrients and oxygen to neurons and glial cells, which would further aggravate local blood volume and flow [61]. In line with these changes, presynaptic release and postsynaptic synaptic plasticity are progressively suppressed in the Katnal2-KO hippocampus. This coincides with the progressive ventriculomegaly in Katnal2-KO mice as well as the observed progressive transcriptomic changes, including synaptic gene down-regulations and ASD-like transcriptomes. These results constitute strong temporal correlations across the molecular/transcriptomic, synaptic, and ventricular phenotypes in Katnal2-KO mice.

Another key question would be whether Katnal2 KO is causally associated with the Katnal2-KO phenotypes. In support of this, early postnatal Katnal2 re-expression driven by intracerebroventricular virus injection prevents the ependymal ciliary enlengthening and ventricular enlargements and partially prevents the courtship USV deficits from occurring in adult Katnal2-KO mice. These results suggest there could be therapeutic potential in normalizing Katnal2 expression at early postnatal stages. This is in line with the emerging evidence indicating the importance of early corrections in animal models of ASD [62].

A previous study on mice lacking Katnal1, which is a Katnal2-related MT-severing protein [63], reported ventricular enlargement, morphological and functional ciliary deficits, and pup USV communication deficits [64]. KATNAL1 mutations in humans have been associated with intellectual disability and microcephaly [65] but minimally with ASD (only 1 ASD-related mutation has been reported thus far) [66]. This contrasts with the large number of ASD-risk KATNAL2 mutations reported to date (49 genetic variants) [22–29]. Therefore, our results extend the notion that MT-related dysfunction can be linked to ependymal/ciliary/ventricle/CSF deficits in ASD, in addition to their impacts on intellectual disability and microcephaly. Our study also suggests that ependymal ciliopathies could underlie ASD, providing new mechanistic insights that expand the previously reported bases for ASD pathophysiology (i.e., synapses, signaling, chromatin, and transcription) [67–70].

Lastly, while revising the manuscript, there was a preprint publication of a similar study on Katnal2-KO mice that display ventriculomegaly [71]. This mutant line lacks exon 8 and is "qualitatively" different from our Katnal2 mouse line that lacks exon 3. Their exon 8 deletion leads to an in-frame deletion in a segment (aa 220–245; 26 aa residues) immediately preceding the AAA-ATPase domain located in the middle of the Katnal2 protein [539 aa-long], likely leaving the exon 8-lacking Katnal2 protein variant largely similar to the whole protein although it might destabilize the protein, which was not directly tested. On the other hand, our exon 3 deletion leads to an out-of-frame protein-truncating mutation in the N-terminal region of the Katnal2 protein at around approximately aa 69, almost completely eliminating the whole Katnal2 protein. The abovementioned preprint publication also reports another Katnal2 mouse line lacking exons 2–12, a larger deletion compared with the exon 8 deletion. However, this mouse line was used to measure ventriculomegaly and choroid plexus-related measures but not other phenotypes that help infer how the ventriculomegaly was induced such as ependymal ciliary functions, making it unclear if the ventriculomegaly observed in these mice involves decreased or increased ependymal ciliary function. These differences and uncertainties should be clarified in future studies.

In summary, our results indicate that deletion of Katnal2, which is highly expressed in ependymal cells, leads to ASD-like social communication deficits that are associated with ventricular enlargements, enhanced ependymal ciliary function, suppressed synaptic function, and ASD-like transcriptomic changes.

## Materials and methods

### Animals

ES cells containing the Katnal2-targeted allele were received from EUCOMM (MAE-4330; Katnal2[tm1a(EUCOMM)Wtsi]) and used to generate transgenic mice. The first-generation mice were backcrossed with mice of C57BL/6J background for more than 5 generations before experiments were conducted. Mating with Protamine-Flp mice was used to generate Katnal2[fl/+] mice. Fertilized eggs (from the breeding with C57BL/6J WT mice) at the two-cell embryo stage were treated with purified HTNC, a cell-permeable Cre recombinase, at a final concentration of 0.3 μm for 30 to 40 min. Katnal2-KO mice were genotyped by polymerase chain reaction (PCR) using the following primer sets: Katnal2 allele: Fwd = 5′-AACAGTGGGAA-CATCACCAGA-3′; Rev = 5′-TCAAACAACCCAGGCACTCA-3′. The expected size of KO PCR band is 259 bp, while WT band is 215 bp. Thy1-EGFP mice were from Jackson Laboratory (Tg(Thy1-EGFP)MJrs/J). All mice were housed and bred at the mouse facility of Korea Advanced Institute of Science and Technology (KAIST) and maintained according to the Animal Research Requirements of KAIST. All animals were fed ad libitum and housed under 12 h light/dark cycle (light phase from 1 AM to 1 PM). Mice were weaned at around the age of postnatal day 21, and mixed-genotype littermate mice of the same sex were housed together until experiments. All animal procedures were approved by the Committee of Animal Research at KAIST (KAIST; KA2020-80) that are adhering to the Animal Protection Act and the Laboratory Animal Act of the Korean government.

### Immunohistochemistry

Mouse brain slices (50 μm; vibratome, Leica) were prepared and stained with DAPI-containing Vectashield (Vector Laboratory) and for Cux1 and FoxP2 using the following commercial antibodies: Cux1 (Santa Cruz sc-13024 at 1:500) and FoxP2 (Abcam ab16046 at 1:500).

### X-gal staining

X-gal staining for Katnal2-β-galactosidase fusion proteins was performed using brain slices (100-μm coronal sections) from Katnal2-KO cassette-containing mice (MAE-4330; Katnal2[tm1a(EUCOMM)Wtsi]; P21 and P56) and X-gal staining (20 mg/ml X-gal; in 2 mM $MgCl_2$, 5 mM $K_4Fe(CN)_6.3H_2O$(Sigma #P-8131), 5 mM $K_3Fe(CN)_6$, 0.01% DOC, 0.02% NP-40 in 1× PBS).

### Combining fluorescence in situ hybridization with immunohistochemistry

Frozen sections (14-μm thick) were cut coronally through the third or fourth cerebral ventricle. Sections were thaw-mounted onto Superfrost Plus Microscope Slides (Fisher Scientific 12-550-15). The sections were fixed in 4% paraformaldehyde (PFA) for 10 min, dehydrated in increasing concentrations of ethanol for 5 min, and finally air-dried. Tissues were then pretreated for protease digestion for 10 min at room temperature. For RNA detection, incubations with different amplifier solutions were performed in a HybEZ hybridization oven (ACDBio) at 40°C. The probe used in this study was 3 synthetic oligonucleotides complementary to the nucleotide (nt) sequence 170–1177 of Mm-Katnal2 (ACDBio). The labeled probe was conjugated to Atto 550 (C1). The sections were hybridized at 40°C with labeled probe per slide for 2

h. Then, the nonspecifically hybridized probe was removed by washing the sections, 3 times each in 1× wash buffer at room temperature for 2 min. Amplification steps involved sequential incubations at 40°C with Amplifier 1-FL for 30 min, Amplifier 2-FL for 15 min, Amplifier 3-FL for 30 min, and Amplifier 4 Alt B-FL for 15 min. Each amplifier solution was removed by washing 3 times with 1× wash buffer for 2 min at RT. Then, sections were postfixed in 4% PFA for 1 h at room temperature and washed with phosphate-buffered saline (PBS). Next, the sections were incubated in primary antibody for 1 h at 37°C. The primary antibodies were diluted in PBS containing 3% BSA and 0.2% Triton X-100. The following primary antibodies were used: acetylated tubulin (Sigma #T6793, 1:500), FoxJ1 (human/mouse, eBioscience #14-9965-82, 1:500), and S100 (Dako #Z0311, 1:500). Next, the sections were washed with PBS and incubated in a cocktail of 488 or 647 conjugated secondary antibodies (Jackson #711-096-152, #715–605–152, and Invitrogen #21202 [1:500]) in PBS containing 3% BSA and 0.2% Triton X-100 for 1 h at room temperature. The secondary antibody was washed with PBS and further incubated with Hoechst (Invitrogen #H3570, 1:1,000) at RT for 10 min. The sections were immersed in mounting solution and images were captured using a confocal microscope (TCS SP8, Leica).

## RNA-Seq analysis

Five mice of 3 and 10 weeks of age were used for each group (WT and KO). The extracted mouse brains were preserved in RNAlater solution (Ambion) and stored at −80°C. Poly-T oligo-attached magnetic beads were utilized to purify poly-A mRNAs. RNA concentrations were quantified using Quant-IT RiboGreen (Invitrogen, R11490), and RNA integrity was determined using TapeStation RNA screen tape (Agilent Technologies), after which only high-quality RNAs (RIN > 7.0) were selected for cDNA library construction using Illumina TruSeq mRNA Sample Prep kit v2 (Illumina). Indexed libraries were submitted to an Illumina HiSeq 4000 (Illumina), and paired-end (2 × 100 bp) sequencing was performed by Macrogen. Transcript abundance was estimated with Salmon (v1.1.0) [72] in Quasi-mapping-based mode onto the Mus musculus genome (GRCm38) with GC bias correction (-gcBias). The acquired abundance data was imported to R (v.3.5.3) with tximport [73] package and differential gene expression analysis was performed using R/Bioconductor DEseq2 (v1.30.1) [74]. Normalized read counts were computed by dividing the raw read counts by size factors and fitted to a negative binomial distribution. The $p$ values were adjusted for multiple testing with the Benjamini–Hochberg correction. Genes with an adjusted $p$ value of less than 0.05 were considered as differentially expressed. GSEA [75,76] was performed to determine whether a priori-defined gene sets would show statistically significant differences in expression between WT and Katnal2-mutant mice. Enrichment analysis was performed using GSEAPreranked (gsea-3.0.jar) module on gene set collections downloaded from Molecular Signature Database (MSigDB) v7.0. GSEAPreranked was applied using the list of all genes expressed, ranked by the fold change, and multiplied by the inverse of the $p$ value with recommended default settings (1,000 permutations and a classic scoring scheme). The false discovery rate (FDR) was estimated to control the false positive finding of a given normalized enrichment score (NES) by comparing the tails of the observed and null distributions derived from 1,000 gene set permutations. The gene sets with an FDR of less than 0.05 were considered as significantly enriched. Integration and visualization of the GSEA results were performed using the EnrichmentMap Cytoscape App (version 3.8.1) [49,77].

## Electrophysiology

Sagittal mouse hippocampal slices (400 and 300 μm thickness for extracellular and intracellular recordings, respectively) were prepared using a vibratome (Leica VT1200) in ice-cold

dissection buffer containing (in mM) 212 sucrose, 25 NaHCO$_3$, 5 KCl, 1.25 NaH$_2$PO$_4$, 0.5 CaCl$_2$, 3.5 MgSO$_4$, 10 D-glucose, 1.25 L-ascorbic acid, and 2 Na-pyruvate bubbled with 95% O$_2$/5% CO$_2$. The slices were recovered at 32˚C for 1 h in normal ACSF (in mM: 125 NaCl, 2.5 KCl, 1.25 NaH$_2$PO$_4$, 25 NaHCO$_3$, 10 glucose, 2.5 CaCl$_2$, and 1.3 MgCl$_2$ oxygenated with 95% O$_2$/5% CO$_2$). For the recording, a single slice was moved to and maintained in a submerged-type chamber at 28˚C, continuously perfused with ACSF (2 ml/min) saturated with 95% O$_2$ and 5% CO$_2$. Stimulation and recording pipettes were pulled from borosilicate glass capillaries (Harvard Apparatus) using a micropipette puller (Narishege).

For extracellular recordings, mouse hippocampal slices (P20-33/juvenile and P56–70/adult) were recorded in the CA1 stratum radiatum region using pipettes filled with ACSF (1 MΩ). fEPSPs were amplified (Multiclamp 700B, Molecular Devices) and digitized (Digidata 1440A, 1550, Molecular Devices) for measurements. The Schaffer collateral pathway was stimulated every 20 s with pipettes filled with ACSF (0.3 to 0.5 MΩ). The stimulation intensity was adjusted to yield a half-maximal response, and 3 successive responses were averaged and expressed relative to the normalized baseline. To induce LTP by theta-burst stimulation (40 trains [4 stimuli/train at 100 Hz with inter-train intervals of 170 ms] separated in 4 bursts [10 trains/burst with inter-burst intervals of 1 s]) after the acquisition of a stable baseline. For input/output recording, a series of increasing input stimuli were given to evoke output signals, after acquiring stable baseline. Fiber volleys and fEPSP slopes that are measured were interpolated by linear fits to plot input/ output signal relationships. The paired-pulse ratio was measured across a range of inter-stimulus intervals of 25, 50, 75, 100, 200, and 300 ms.

Whole-cell patch-clamp recordings of hippocampal CA1 pyramidal neurons were made using a MultiClamp 700B amplifier (Molecular Devices) and Digidata 1440A, 1550 (Molecular Devices). During whole-cell patch-clamp recordings, the series resistance was monitored for each sweep by measuring the peak amplitude of the capacitance currents in response to short hyperpolarizing step pulse (5 mV, 40 ms); only cells with a change in <20% were included in the analysis. To measure NMDA/AMPA ratio, mouse hippocampal slices (P20–21/P56–70) were used. The recording pipettes (2.5 to 3.5 MΩ) were filled with an internal solution containing the following (in mM): 100 CsMeSO$_4$, 10 TEA-Cl, 8 NaCl, 10 HEPES, 5 QX-314-Cl, 2 Mg-ATP, 0.3 Na-GTP and 10 EGTA, with pH 7.25, 295 mOsm. CA1 pyramidal neurons were voltage clamped at −70 mV, and EPSCs were evoked at every 15 s. AMPAR-mediated EPSCs were recorded at −70 mV, and 20 consecutive responses were recorded after stable baseline. After recording AMPAR-mediated EPSCs, holding potential was changed to +40 mV to record NMDAR-mediated EPSCs. NMDA component was measured at 60 ms after the stimulation. The NMDA/AMPA ratio was determined by dividing the mean value of 20 NMDA components of EPSCs by the mean value of 20 AMPAR-mediated EPSC peak amplitudes. Decaying tau of NMDA-R EPSC was measured since 60 ms after the stimulus to rule out intervening AMPA current. To measure excitability of hippocampal CA1 cells (P20–21/P56–70), recording pipettes (2.5 to 3.5 MΩ) were filled with an internal solution containing the following (in mM): 137 K-gluconate, 5 KCl, 10 HEPES, 0.2 EGTA, 10 Na-phosphocreatine, 4 Mg-ATP, and 0.5 Na-GTP, with pH 7.2, 280 mOsm. To inhibit postsynaptic responses, picrotoxin (100 μm), NBQX (10 μm), and D-AP5 (50 μm) were added. After rupturing the cell, the currents were clamped, and RMP was measured. Cells with RMP larger than −60 mV were not used. After stabilizing cell, RMP was adjusted by −65 mV. Current input was increased from 0 to 360 in increments of 30 pA per sweep. Each current was injected with a time interval of 15 s.

Data were acquired by Clampex 10.2 (Molecular Devices) and analyzed by Clampfit 10 (Molecular Devices). Drugs were purchased from Abcam (TTX), Tocris (NBQX, D-AP5), and Sigma (picrotoxin, DCS).

## Electron microscopy

For SEM, brain cortical tissues were harvested and fixed in 1% glutaraldehyde and 2% OsO$_4$ solution. After washing in 0.1 M sodium phosphate buffer (pH 7.2), samples were dehydrated using acetone in a sequential gradient and through the critical point dryer (Leica CPD300). After mounting, samples were coated with gold particles and observed on a field emission scanning electron microscope (Hitachi Regulus 8100). Adobe Photoshop was used for the analysis of ependymal ciliary length.

## Magnetic resonance imaging

All MRI experiments were performed on a Bruker Biospec 9.4 T/30 cm horizontal bore instrument with an actively shielded 12.0 cm diameter insert. Mice were initially anesthetized using 2% to 5% isoflurane. Then, the mouse was positioned in an MR cradle with a bite bar and ear bars, and maintained with 1.0% to 1.2% isoflurane at 37 ± 0.5˚C using a circulating warm water blanket throughout MRI studies. A homogenous coil (86 mm inner diameter) was used for excitation, and an actively decoupled Bruker planar surface coil (10 mm inner diameter) positioned on top of the mouse head was used for detection.

For brain volume analysis, anatomical images were acquired using a T2 RARE sequence with a RARE factor of 4. Four averages of 20 axial slices were acquired with a field of view of $15 \times 7.5$ mm, a spatial resolution of $50 \times 50 \times 500$ μm$^3$, matrix size of $256 \times 256$, effective echo time (TE) of 30 ms, echo spacing of 15 ms, and recovery time (TR) of 4 s. The Allen mouse brain atlas (https://atlas.brain-map.org/) was spatially registered to individual anatomical images with Advanced Normalization Tools (ANTs) (http://stnava.github.io/ANTs/) [78]. The bilateral lateral ventricles were visually identified from anatomical images in each slice, and their volumes were determined. Also, regions of interest were identified, based on the Allen brain atlas, and their volumes were calculated from each 3D anatomical image in each animal.

For CSF flow analysis, 3D phase-contrast MRI with velocity encoding (FLOWMAP) was used for mapping CSF velocity. A velocity encoding coefficient of 4 cm/s was used in the slice direction. Sixteen averages of 20 axial slices were acquired with a resolution of $156 \times 156 \times 500$ μm$^3$, TE = 5 ms, TR = 17.461 ms, and flip angle = 20˚. The flow images were analyzed with ParaVision 6.0.1 software (Bruker Biospin). The cerebral aqueduct appearing around in the midline brain regions was identified by visual inspections of anatomical images and flow maps as well as the brain atlas. Mean CSF velocity was determined in the $2 \times 2$ pixel ROI of the cerebral aqueduct.

For cerebral blood volume measurements [44], T$_2$*-weighted single-shot GE-EPI were acquired with the following parameters: TR/ TE = 1,000/11.6 ms, flip angle = 50˚, spatial resolution = $156 \times 156 \times 500$ μm$^3$, and 20 axial slices. Transient hypoxia was induced by reducing the fraction of inspired oxygen (FiO2) from approximately 50% to 0% with a three-channel programmable gas mixer (GSM-3 Gas Mixer, CWE) synchronized with the MRI scanner. A 150-s run consisted of 20-s-normoxia, 5-s anoxia, 60-s normoxia, 5-s anoxia, and 60-s normoxia. Three to 5 runs were repeated and averaged. The Allen mouse brain atlas (https://atlas.brain-map.org/) was spatially registered to individual anatomical images with Advanced Normalization Tools (ANTs) (http://stnava.github.io/ANTs/) [78]. Thus, areas under the curve (AUC) over 20 s following the onset of 5-s stimulus were calculated at a voxel or ROI basis. The AUCs were normalized by an average across AUCs of vessel voxels (2 highest AUC voxels in each animal). Normalized AUCs are directly related to cerebral blood volume fraction (% blood volume).

## Beads flow assay

Ependymal flow analysis using fluorescent microbeads was performed as described [79]. Briefly, mice were sacrificed, and lateral walls (LWs) were exposed by removing the

hippocampus, cortex, thalamus, and choroid plexus in Leibovitz L-15 medium at 37°C. Then, the LWs were immobilized on a dissecting dish using 2 pins in fresh L-15 medium. A grinded glass microcapillary filled with FluoSpheres microbeads (2.0 mm diameter, Invitrogen, F8827) in 5% glycerol solution was positioned just above the dorsal-medial surface around the adhesion area of the LWs. Micro bead solution was carefully ejected using a Cell Tram Oil micromanipulator (Eppendorf), and the movements of microbeads were recorded in 200 msec time intervals through ZEISS Axio Zoom.V16 fluorescent dissection microscope. High-speed videos of posterior-medial microbead flow were exported through ZEN Blue imaging software at 5.0 frames per second (fps). The speed of bead movement was measured using a manual tracking method in ImageJ program (fiji-win64). Pneumolysin (0.5 μg/ml) was treated in the L-15 media for 10 min prior to the experiment.

## Ciliary beating frequency

Coronal mouse lateral ventricle slices (250 μm thickness) were prepared using a vibratome (Leica VT1200) in ice-cold dissection buffer containing (in mM) 212 sucrose, 25 $NaHCO_3$, 5 KCl, 1.25 $NaH_2PO_4$, 0.5 $CaCl_2$, 3.5 $MgSO_4$, 10 D-glucose, 1.25 L-ascorbic acid, and 2 Na-pyruvate bubbled with 95% $O_2$/5% $CO_2$. The slices were recovered at 32°C for 20 min in normal ACSF (in mM: 125 NaCl, 2.5 KCl, 1.25 $NaH_2PO_4$, 25 $NaHCO_3$, 10 glucose, 2.5 $CaCl_2$, and 1.3 $MgCl_2$ oxygenated with 95% O2/5% CO2). Prepared samples were imaged using Nikon Ti2 inverted microscope equipped with 60× objective lens (Nikon, Plan Apo 60× Oil). A microscope incubator (OKOLAB, H301-NIKON-TI-S-ER) was installed on the microscope to keep the sample at 37°C and 5% $CO_2$. Live movies of beating cilia were acquired at the speed of approximately 1,000 fps using a high-speed camera (PCO, pco.dimax cs4).

To quantify ciliary beating frequency, 16~20 ROIs (11 × 11 pixels each; at least 30 pixels apart) were confined to the ependymal layers for each video. In each ROI, the brightness change was Fourier transformed to determine ciliary beating frequency. The custom MATLAB code used for the analysis is available in https://github.com/sjleen/Cilia-stroke-analysis.

## Intracranial pressure measurement

To measure intracranial pressure (ICP), mice were anesthetized with ketamine/xylazine and placed in a stereotaxic frame. The cisterna magna was exposed surgically, and a pressure transducer (FISO 75–0706) in an ACSF-filled glass-pulled capillary tube was positioned outside the membrane of the cisterna magna. A reference measurement was taken externally within the applied ACSF, and then the sensor was advanced into the cisterna magna. Pressure was recorded for a further 60 s using FISO Evolution software (v2.2.0.0). Traces were evaluated for patency, stability, and the presence of cardiorespiratory ICP waveforms and processed in R to calculate mean ICP during the measurement period.

## Katnal2 re-expression

To generate the Katnal2-overexpressing AAV construct (AAV-CAG-mKatnal2--p2A-EGFP-WPRE; WPRE for translational increase), the Myc-DDK/Flag-tagged mouse Katnal2 cDNA (Origene #MR220406L1) construct was used to subclone mouse Katnal2 cDNA into the AAV-CAG-GFP (Addgene #28014) vector. Here, some C-terminal aa sequence (aa 406–539) missing in the Origene mouse Katanal2 clone and the p2A sequence (GSGATNFSLLKQAGDVEENPGP) were synthesized using overhang-PCR. AAV particles were prepared using HEK293T cells transfected with the target plasmid clone (abovementioned), PHP.eB plasmid (a kind gift from Dr. Viviana Gradinaru), and pAAV-helper. For virus injection into the lateral ventricles of the mouse brain, the head of mice (P8–12) were

fixed on stereotaxic apparatus (Kopf Instruments) under isoflurane (Piramal Healthcare) anesthesia. An injection needle containing virus solutions was moved down at the speed of 1 mm/min, and the solution was injected into the brain at the speed of 160 nL/min, 500 nL in total. Targets locations of the left and right ventricles (for bilateral injections) was DV (−2.0 mm), ML (−0.8/+1.0), and AP (+3.4) from the lambda on the skull (lambda was used instead of bregma because of the young age of the mouse). Their bregma-lambda length at P8–12 was 3.6 mm, shorter than that in adult mice (4.2 mm). As a negative control for AAV-CAG-mKatnal2-EGFP-WPRE, AAV-GFAP-EGFP was bilaterally injected. Both Katnal2-overexpressing AAV construct and control virus were bilaterally injected 0.5 μl each.

## Immunoblot

Expression levels of Katnal2 proteins were determined by immunoblot analyses. Because Katnal2 protein levels in the brain were below the detection limit of immunoblotting using the currently available homemade Katnal2 antibodies (#2167; guinea pig polyclonal), which targeted the last 30 aa (aa 517–539) of the mouse Katnal2 protein, we used testis samples (8 weeks) prepared by tissue homogenization (150 mM NaCl, 1 mM EDTA, 50 mM Tris-HCl [pH = 7.4], 1% Triton-X100) and centrifugation followed by the use of the supernatant (not pellet).

For immunoblot analyses of choroid plexus lysates, 4 choroid plexus samples from 2 mice were pooled to make *n* of 1 and homogenized in RIPA buffer (10 mM HEPES, 150 mM NaCl, 1 mM EDTA, 0.1 mM MgCl2, 1% NP-40 with protease inhibitor). The following antibodies were utilized for immunoblotting: aquaporin-1 (Sigma-Aldrich, AB2219; 1:1,000), Na$^+$/K$^+$-ATPase α1 (Sigma-Aldrich, 05–369; 1:1,000), AE2 (Santa Cruz; sc-376632; 1:1,000), and GAPDH (Cell Signaling, D4C6R; 1:1,000). Full-length immunoblot images can be found in **S1 Raw Images**.

## Behavioral experiments

Male mice were used for the behavioral experiments.

## Ultrasonic vocalization

An ultrasound microphone (Avisoft) and Avisoft Recorder software were used to record mouse USVs, a form of social communication in rodents [80]. For courtship USVs, a subject male mouse was placed in a new home cage with an age-matched unfamiliar C57BL/6J female mouse, followed by USV recordings for 5 min. Recorded USVs were analyzed as previously described [81]. Briefly, Avisoft SASLab Pro software (RRID:SCR_014438) was used to analyze USVs. Signals were filtered from 1 to 100 kHz and digitized with a sampling frequency of 250 kHz and 16 bits per sample (Avisoft UltraSoundGate 116H). To generate spectrograms, the following parameters were used: FFT length: 256, frame size: 100, window: FlatTop, overlap: 75%, resulting in a frequency resolution of 977 Hz and a temporal resolution of 0.256 ms. USV frequencies lower than 45 kHz were filtered out to reduce background white noises.

## Three-chamber test

The size of the three-chambered apparatus was 40 cm W × 20 cm H × 26 cm D with a center chamber of 12 cm W and side chambers of 14 cm W. In the first session, a subject mouse could freely move around the whole three-chambered apparatus with 2 small plastic containers in the left or right corners for 10 min (Session 1). The mouse was then gently confined in the center chamber while a novel "Object" and a WT stranger mouse "Stranger 1 (129Sv strain)"

was placed in the 2 plastic containers. The subject mouse was then allowed to freely explore all 3 chambers for 10 min (Session 2). In the third session, the subject mouse was again gently guided to the center chamber while the "Object" was replaced with a WT "Stranger 2" mouse. The subject mouse again freely explored all 3 chambers for 10 min (Session 3). Object/Stranger exploration was defined by the mouse's nose being oriented toward the target and coming within 2 cm of it as measured by EthoVision XT 12 program (Noldus).

### Direct interaction test

For the direct interaction test between mice in the same genetic background (C57BL/6J), each mouse was first habituated in a direct social interaction box for 30 min on the day before experiment. On the test day, pairs of mice of the same ages and genotypes, which have not met before, were placed in a direct interaction box, and their interactions were recorded for 10 min. For the direct interaction test between mice in different genetic backgrounds (C57BL/6J and C3H/HeJ), WT or Katnal2-KO males in the C57BL/6J background were paired with age/weight-matched non-familiar C3H/HeJ males (Jackson Labs). Each subject mouse and C3H/HeJ partner were placed in the direct social interaction box and allowed to freely interact for 10 min. Social interactions were recorded using a top-view camera and quantified manually by trained researchers in a blind manner using the Premiere Pro 2020 program (Adobe). The behaviors from the unidirectional social interaction test were grouped into 2 categories: reciprocal social interaction where one mouse socially explores the other and the other animal reciprocates the social interaction; nonreciprocal interaction where social actions of Katnal2-KO mouse were not reciprocated by the C3H/HeJ partner.

### Open-field test

Mice were placed in an open-field box (40 × 40 × 40 cm) and recorded with a video camera for 60 min. The center zone line was 10 cm apart from the edge. The testing room was illuminated at approximately 0 lux. Mice movements were analyzed using EthoVision XT 12 program (Noldus).

### Rotarod test

Mice were placed on the rotating rod for 10 s, followed by the start of rod rotation. The rotating speed of the rod was gradually increased from 4 to 40 rpm over 5 min. The assay was performed for 5 consecutive days while measuring the latencies of mice falling from the rod or showing 360-degree rotation on the rod.

### Elevated plus maze

The elevated plus maze consisted of 2 open arms, 2 closed arms, and a center zone, and was elevated to a height of 50 cm above the floor. Mice were placed in the center zone and allowed to explore the space for 8 min. The data was manually quantified by trained researchers in a blind manner.

### Light-dark chamber test

The light-dark test apparatus consisted of light (approximately 400 lux) and dark (approximately 0 lux) chambers that adhered to each other. The size of the light chamber was 20 × 30 × 20 cm, and that of the dark chamber was 20 × 13 × 20 cm. An entrance enabled mice to freely move across the light and dark chambers. Mice were introduced to the center of the light chamber and allowed to explore the apparatus freely for 5 min. The time spent in dark and

light chambers and transition frequencies were measured using EthoVision XT 10 program (Noldus).

### Novel object recognition test

The novel object recognition test was performed in the open-field box. On the day after the open-field test, mice were allowed to freely explore 2 identical objects (blue cylinder or silver-colored box) in the open-field box for 20 min. Twenty-four hours later, mice were placed in the same box where one of the 2 objects was replaced with a new one. Sniffing time for each object was measured. Object exploration was defined by the mouse's nose being oriented toward the object and coming within 2 cm of it as measured by EthoVision XT 12 program (Noldus).

### Morris water maze

Mice were trained to find the hidden platform (10-cm diameter) in a white plastic tank (120-cm diameter). Mice were given 3 trials per day with an inter-trial interval of 30 min. The learning-phase experiments of the water maze were performed for 7 consecutive days, followed by the probe test on day 8 where mice were given 1 min to find the removed platform. For reversal training (days 9 to 13), the location of the platform was switched to the opposite position from the previously trained quadrant, and mice were trained to learn the new position of the platform. Target quadrant occupancy and the exact number of crossings over the former platform location during the probe test were measured using EthoVision XT 12 program (Noldus).

### Acoustic startle

Acoustic startle responses were measured to determine auditory functions. After 5 minutes of 65-dB background acclimation in the acoustic startle apparatus (San Diego Instrument; SR-Lab ABS system #2325–0400), 92 of 40-ms acoustic stimulus (7 each for 70~120 dB with 5-dB intervals; 7 mock trials; 8 of 120 dB trials located 4 each at the beginning and the end) were given to subject mice in pseudorandomized order. The startle responses were recorded and analyzed using the SR-LAB software (San Diego Instrument).

### Contextual fear conditioning test

Subject mice were introduced to the fear-conditioning box (Coulbourn Instrument; H13-16) for 10 min and allowed to freely explore the box a day before the day of the experiment. On the conditioning day, mice were placed in the same box (20 lux; grid floor). After 2 min of acclimation, 3 electric foot shocks (0.5 mA, 2 s) were delivered at 1 min intervals. After 3 min of the post-shock period, mice were returned to home cages. Twenty-four hours after, mice were placed in the fear box with the same condition to monitor behaviors for 3 min. This monitoring experiment was repeated 7 days after the conditioning. A top-view camera was to record mouse movements, which were analyzed using the ActiMetrics FreezeFrame 3 software (freezing threshold = 10%).

### Immunohistochemistry

Mouse brain slices (50 μm; vibratome, Leica) were prepared and stained with DAPI-containing Vectashield (Vector Laboratory) and for Cux1 and FoxP2 using the following commercial antibodies: Cux1 (Santa Cruz sc-13024 at 1:500) and FoxP2 (Abcam ab16046 at 1:500).

## Immunocytochemistry

Cultured hippocampal neurons (DIV 7) were fixed in 4% PFA/4% sucrose-containing Tyrode's solution (119 mM NaCl, 2.5 mM KCl, 2 mM $CaCl_2$, 2 mM $MgCl_2$, 25 mM HEPES, 30 mM glucose (pH 7.4)) for 15 min, permeabilized in 0.25% Triton X-100 in Tyrode's solution for 5 min, and then blocked in 5% Normal Donkey Serum (NDS)-containing Tyrode's solution at 37˚C for 30 min. Then, tissues were treated with primary antibodies (anti-Arl13b (Proteintech 17711-1-AP, 1:1000); anti-NeuN (EMD Millipore ABN90, 1:1,000)) containing Tyrode's solution with 5% NDS at 37˚C for 2 h and secondary antibodies (1:1,000) for 45 min. The length of primary cilia was measured by ImageJ software (fiji-win64).

## X-gal staining

X-gal staining for Katnal2-β-galactosidase fusion proteins was performed using brain slices (100-μm coronal sections) from Katnal2-KO cassette-containing mice (MAE-4330; Katnal2^tm1a(EUCOMM)Wtsi^; P21 and P56) and X-gal staining (20 mg/ml X-gal; in 2 mM MgCl2, 5 mM K4Fe(CN)6.3H2O(Sigma #P-8131), 5 mM K3Fe(CN)6, 0.01% DOC, 0.02% NP-40 in 1× PBS).

## CSF drainage assay

To check the CSF drainage to the deep cervical lymph node, the head of anesthetized mice was fixed on the stereotaxic frame, and 1 μl of Alexa Fluor 488-conjugated ovalbumin (Invitrogen O34781) was injected for 5 min in the lateral ventricle. The detailed coordinate is the following: AP = 0.0; ML = −1.2; DV = −2.4 from Bregma. After 3-min break, the injection needle was removed. After recovery for an hour on the 37˚C heating pad, injected mice were cardiac perfused with Heparin-containing PBS/PFA, dCLNs were dissected out and cryoprotected in 15%/30% serial. The lymph nodes were prepared into 40-μm sections by using cryostat, and costained with DAPI, and imaged using LSM-780 (Zeiss). The coverage of OVA-488 and DAPI of the lymph nodes were measured by ImageJ program (fiji-win64).

## Statistics

For group comparisons, Student's *t* test or Mann–Whitney test was used based on the results of normality testing using D'Agostino & Pearson or Shapiro–Wilk normality test. The Mann–Whitney test was used for any column that yielded a *p* value < 0.05 in either normality test. For multi-comparisons with 2 independent variables, two-way ANOVA and Sidak's multiple comparison test were used. For relative brain regional volumes/CBV, adjusted *p* values from one-sample *t* test were used. For ciliary beating frequencies, we used a permutation test with 10,000 iterations (MATLAB). For the comparison of 3 groups with the 1 independent variable (rescue experiment), we used one-way ANOVA with Dunnet's post hoc test. GraphPad Prism 7 was used for statistical analyses. See **S1 Data** for statistical details.

## Supporting information

**S1 Fig. Gene knockout strategy for Katnal2-KO mice.** (A) Domain structure of the Katnal2 protein (539 aa-long), known sites of ASD patient-derived point mutations (49 variants from 14 reports), and the protein regions corresponding to exon 3, which was deleted in Katnal2-KO mice (55 aa in the LisH domain). LisH domain, lissencephaly-1 homology domain; AAA-ATPase domain, AAA family ATPase domain. (B) Katnal2 knockout (KO) strategy in mice. The same primer sets are used to detect WT and mutant PCR products (259 and 215 base pairs); they differ by the presence or absence of exon 3, deletion of which leads to a shift

in the open-reading frame. (C) PCR genotyping for WT, heterozygous Katnal2-KO (HT), and homozygous Katnal2-KO (KO) mice (postnatal day [P56]). (D) Validation of Katnal2 KO by immunoblot analysis; we used mouse testis samples (P56) instead of brain samples because the expression levels of Katnal2 protein are much greater in the testis relative to the brain. In the brain, Katnal2 expression is confined to select brain regions and cell types, such as ependymal cells lining ventricular walls, as shown by X-gal staining (see **Figs 2A and S5**). Note also that there is a major Katnal2 protein band (approximately 45 KDa; arrowhead) in WT but not KO testis samples, as revealed by immunostaining with Katnal2 polyclonal antibodies (#2817) raised against aa 517–539 of the Katnal2 protein. Asterisk indicates a nonspecific band recognized by Katnal2 antibodies. (E) Normal body weights in Katnal2-KO mice (2 months). ($n$ = 7 mice [WT], 9 mice [KO], Student's $t$ test). (F and G) Normal superficial (layer 2/3) and deep (layer 6) cortical layer structures in the Katnal2-KO brain (P56), as revealed by double immunofluorescence staining for NeuN (a neuronal marker) and Cux1 (layer 2/3 marker) or for NeuN and FoxP2 (layer 6). DAPI staining was performed for nuclear staining. The examples shown here are from the somatosensory cortex (layers 1–6). (H) Sholl analysis of CA1 hippocampal neurons from WT and Katnal2-KO mice, visualized by cross-breeding with Thy1-EGFP mice. (P21–23; $n$ = 19 cells from 6 [WT], 19, 6 [KO], two-way RM-ANOVA). (I) Comparable lengths of the primary cilia in cultured hippocampal neurons (days in vitro 7) from WT and Katnal2-KO embryos. ($n$ = 147 images from 11 glass slides from 4 mice [WT], 165, 11, 4 [KO], Student's $t$ test). Data values represent means ± SEM. Significance is indicated as ns (not significant). Statistical results and numerical data values can be found in **S1 Data**. Full-length PCR and immunoblot images can be found in **S1 Raw Images**.
(TIF)

**S2 Fig. Abnormal social communication and mounting in Katnal2-KO mice.** (A) Normal levels of social approach and social novelty recognition in Katnal2-KO mice (2–3 months; male) in the three-chamber test, as shown by time spent sniffing social and object targets (S1/S2, old/new social target; O, object), time spent in the chamber, and the preference index (time difference for S1–O [S2 –S1] / total time × 100). ($n$ = 17 mice [WT], 16 [KO], two-way RM-AVOVA [sniffing time, chamber time], Student's $t$-test [sociability index, social novelty recognition index]). (B) Normal levels of direct social interaction in Katnal2-KO mice (2–3 months) in the direct social interaction test, wherein freely moving WT/mutant mouse pairs were used to measure nose-to-nose, nose-to-tail, and following. ($n$ = 11 pairs [WT], 13 [KO], two-way RM-ANOVA). (C) Normal levels of social interaction in Katnal2-KO mice (2–3 months) in a modified version of the direct social interaction test, where a subject mouse interacted with a stranger mouse of C3H background (i.e., with a different coat color) and unidirectional and bidirectional/reciprocal social interactions were measured. ($n$ = 10 mice [WT], 11 [KO], two-way RM-ANOVA). (D) Increased courtship USVs upon encountering a novel female stranger mouse is seen for Katnal2-KO male mice (2–3 months), as indicated by USV call frequency and duration (total and each call). ($n$ = 19 [WT], 22 [KO], Student's $t$ test [frequency, each-call duration], Mann–Whitney test [total duration]). (E) Largely normal social interactions during the courtship tests, except for mounting, as shown by analysis of male-to-female social interactions (nose-to-nose, nose-to-tail, following), female-to-male interactions (nose-to-nose, nose-to-tail, following), bidirectional interactions (dyadic), and mounting. Mounting behaviors alone show a genotype-related difference, as determined by Student's $t$ test. Two-way ANOVA encompassing all behaviors did not reveal a genotype-related difference. ($n$ = 12 [WT], 14 [KO], two-way RM-ANOVA). (F) Decreased mounting success in Katnal2-KO male mice (2–3 months), as shown by the proportion of mice that succeeded in mounting during a courtship-test session (5 min). ($n$ = 15 [WT], 20 [KO], Chi-square test). (G) Normal repetitive self-

grooming in Katnal2-KO mice (2–3 months) in a new home cage without bedding. ($n = 13$ [WT], 17 [KO], Student's *t* test). Data values represent means ± SEM. Significance is indicated as * ($<0.05$), ** ($<0.01$), *** ($<0.001$) or ns (not significant). Statistical results and numerical data values can be found in **S1 Data**.
(TIF)

**S3 Fig. Normal levels of locomotion, anxiety-like behavior, acoustic startle, motor coordination, and learning and memory in Katnal2-KO mice.** (A) Normal levels of locomotor activity in the open-field test for Katnal2-KO mice (2–3 months), as shown by distance moved. Note that there is no genotype-related difference in the time spent in the center region of the open-field area, suggestive of normal anxiety-like behavior in the mutant mice. ($n = 18$ mice [WT], 21 [KO], two-way RM-ANOVA [distance moved and time in center], Mann–Whitney test [total distance moved], Student's *t* test [total time in center]). (B) Normal levels of anxiety-like behavior in the elevated plus-maze for Katnal2-KO mice (2–3 months), as shown by % time spent in open arms. ($n = 18$ [WT], 22 [KO], Student's *t* test). (C) Normal levels of anxiety-like behavior in the light-dark test for Katnal2-KO mice (2–3 months), as shown by time in the light box. ($n = 12$ [WT], 16 [KO], Student's *t* test). (D) Normal levels of acoustic startle for Katnal2-KO mice (2–3 months). ($n = 9$ [WT], 7 [KO], two-way RM-ANOVA). (E) Normal levels of motor coordination and learning in the rotarod test in Katnal2-KO mice (2–3 months), as shown by the latency to fall. ($n = 17$ [WT], 15 [KO], two-way RM-ANOVA). (F) Normal levels of learning and memory in the forward and reversal phases of the Morris water maze test in Katnal2-KO mice (2–3 months). ($n = 12$ [WT], 15 [KO], two-way RM-ANOVA [escape latency, time in quadrant in probe test and reverse probe test], Mann–Whitney test [crossing number], Student's *t* test [crossing number (reversal)]). (G) Normal levels of object recognition memory in the novel object recognition test in Katnal2-KO mice (2–3 months), as shown by the novel-object preference (% time spent exploring the novel object). ($n = 13$ [WT], 10 [KO], Student's *t* test). (H) Normal levels of learning and memory in the contextual fear memory test in Katnal2-KO mice (2–3 months), as shown by freezing levels during fear acquisition, at 24-h retrieval, and at subsequent 8-day retrieval. ($n = 18$ [WT], 15 [KO], two-way RM-ANOVA). Data values represent means ± SEM. Significance is indicated as ns (not significant). Statistical results and numerical data values can be found in **S1 Data**.
(TIF)

**S4 Fig. Moderately increased ventricular areas in P7 Katnal2-KO mice and largely normal intracranial brain volumes in P70 Katnal2-KO mice.** (A) Moderately increased areas of lateral ventricles in Katnal2-KO mice at P7, as shown by measurements derived from coronal brain slices. Note that brain areas are not increased at P7. AP axis, anterior-posterior axis. Scale bar, 2 mm. ($n = 5$ mice [WT], 6 [KO], two-way RM-ANOVA). (B) Largely normal intracranial brain volumes, with very moderate increases in select brain regions, among Katnal2-KO mice (3 months), as shown by MRI analyses of absolute and relative brain volumes (two-way ANOVA). Note, however, that there are moderate Katnal2-KO-dependent increases some brain regions (one-sample *t* test). ($n = 8$ mice [WT], 6 [KO], two-way RM-ANOVA and one-sample *t* test). Significance is indicated as * ($<0.05$) or ns (not significant). Statistical results and numerical data values can be found in **S1 Data**.
(TIF)

**S5 Fig. Distribution patterns of Katnal2 proteins in the mouse brain, as revealed by X-gal staining.** (A and B) Distribution patterns of Katnal2 proteins in the mouse brain, as revealed by X-gal staining of Katnal2-β-galactosidase fusion proteins expressed in Katnal2-KO mice

(P21 and P56) with the β-geo cassette unremoved. Scale bar, 1 mm.
(TIF)

**S6 Fig. Additional examples of SEM images used to determine ependymal ciliary length.** (A and B) Additional examples of SEM images from the lateral ventricles of WT and Katnal2-KO mice (P28-33; 6 images from 6 WT mice and 11 images from 11 KO mice). (TIF)

**S7 Fig. Comparison of CBVs in different brain regions of WT and Katnal2-KO mice.** (A) Comparisons of cerebral blood volumes (CBVs) in WT and Katnal2-KO brains, as shown by the extents of decreases in fMRI signals induced by hypoxic nitrogen stimulus in different brain regions (left, relative CBVs) and the KO values normalized to WT values (right, %WT). Iso-cortex includes somatomotor area, somatosensory area, gustatory area, visceral area, auditory area, visual area, anterior cingulate area, prelimbic area, infralimbic area, orbital area, agranular insular area, retrosplenial area, posterior parietal association, temporal association area, perirhinal area, and ectorhinal area. ($n$ = 8 mice [WT], 6 [KO], two-way ANOVA and one-sample $t$ test). (B) Comparable levels of relative CBVs (BOLD signals) in lateral ventricles containing the choroid plexus region in WT and Katnal2-KO mice, as assessed by hypoxic nitrogen stimulus. The red-colored areas indicate representative choroid plexus-containing voxels that we used for the signal tracing. ($n$ = 9 mice [WT], 9 [KO], Mann–Whitney test). (C) Minimal levels of relative CBVs (BOLD signals) in lateral ventricles without the choroid plexus region in WT and Katnal2-KO mice, indicating that the majority of BOLD signals in lateral ventricles are from the choroid plexus and that the effect of the CSF-containing region on BOLD signals is minimal. The yellow-colored areas indicate representative choroid plexus-non-containing lateral ventricle voxels that we used for the signal tracing. (D) Lack of genotype differences in the levels of ion co-transporters (Anion Exchanger 2/AE2 and Na$^+$/K$^+$-ATPase subunit α1) and water channels (aquaporin-1/AQP-1 Student's $t$-test), as shown by immunoblot analysis of choroid plexus lysates from WT and Katnal2-KO mice ($n$ = 6 samples [WT], 5 [KO]; 4 choroid plexus samples from 2 mice were pooled to make $n$ of 1). (E) Experimental scheme for the measurement of CSF drainage. Fluorescent ovalbumin proteins (OVA-488) were introduced by intracerebroventricular (icv) injection into the lateral ventricles of WT and Katnal2-KO mice (2 months), and signals were measured in the deep cervical lymph nodes (dCLNs). (F) Example images and quantification of fluorescent ovalbumin proteins detected in the dCLNs outside the brain. DAPI staining was performed to label cells. Scale bar, 100 μm. ($n$ = 3 mice [WT], 5 [KO], Mann–Whitney test). Data values represent means ± SEM. Significance is indicated as * ($<$0.05), ** ($<$0.01), or ns (not significant). Statistical results and numerical data values can be found in **S1 Data**. Full-length immunoblot images can be found in **S1 Raw Images**.
(TIF)

**S8 Fig. Volcano plots of DEGs for P21-Katnal2/WT and P70-Katnal2 transcripts.** (A and B) Volcano plots of differentially expressed genes (DEGs) from P21- and P70-Katnal2/WT transcripts, shown together with the lists of top-ranked DEGs, sorted by $p$ values. ($n$ = 5 mice for WT and KO, FDR $<$ 0.05 and fold-change $>$ 1.5-fold). Detailed DEG results can be found in **S3 Data**.
(TIF)

**S1 Data. Statistical details and numerical values.**
(XLSX)

**S2 Data. Total RNA-Seq results.**
(XLSX)

**S3 Data. DEGs from P21-Katnal2/WT and P70-Katnal2 transcripts.**
(XLSX)

**S4 Data. GSEA results for P21-Katnal2/WT and P70-Katnal2 transcripts.**
(XLSX)

**S5 Data. Lists of gene sets and genes in each gene sets used in the present study.**
(XLSX)

**S1 Raw Images. Full-length images of immunoblotting and PCR experiments.**
(PPTX)

**S1 Movie. An example of ependymal ciliary beating movements.** An example of ependymal ciliary beatings observed in lateral ventricles of WT mice (P28-42). The movie was generated using images acquired at approximately 1,000 frames/sec.
(MP4)

## Acknowledgments

We would like to thank Dr. Ji-Seon Seo for helpful comments, the facilities and the scientific and technical assistance of the EM & Histology Core Facility, and Dr. Yongsuk Hur at the Bio-Medical Research Center, KAIST.

## Author Contributions

**Conceptualization:** Ryeonghwa Kang, Kyungdeok Kim, Eunjoon Kim.

**Data curation:** Ryeonghwa Kang, Kyungdeok Kim, Miram Shin, Kwangmin Ryu, Subin Choi, Esther Yang, Wangyong Shin, Seungjoon Lee, Suho Lee, Zachary Papadopoulos, Ji Hoon Ahn.

**Formal analysis:** Ryeonghwa Kang, Kyungdeok Kim, Yewon Jung, Hyojin Kang.

**Funding acquisition:** Eunjoon Kim.

**Investigation:** Ryeonghwa Kang, Kyungdeok Kim, Yewon Jung, Sang-Han Choi, Chanhee Lee, Geun Ho Im, Miram Shin, Kwangmin Ryu, Subin Choi, Esther Yang, Wangyong Shin, Ji Hoon Ahn.

**Methodology:** Seungjoon Lee.

**Supervision:** Gou Young Koh, Hyun Kim, Won-Ki Cho, Soochul Park, Seong-Gi Kim, Eunjoon Kim.

**Visualization:** Ryeonghwa Kang, Kyungdeok Kim.

**Writing – original draft:** Ryeonghwa Kang, Kyungdeok Kim, Eunjoon Kim.

**Writing – review & editing:** Gou Young Koh, Jonathan Kipnis, Hyun Kim, Won-Ki Cho, Soochul Park, Seong-Gi Kim.

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
