## [Editor Report · Decision Letter 0]

17 Mar 2023

Dear Eunjoon, 

Thank you for submitting your manuscript entitled "Ependymal ciliary hyperfunction in Katnal2-mutant mice with hydrocephalus and ASD-related social, synaptic, and transcriptomic deficits" for consideration as a Research Article by PLOS Biology.

Your manuscript has now been evaluated by the PLOS Biology editorial staff as well as by an academic editor with relevant expertise and I am writing to let you know that we would like to send your submission out for external peer review.

Once your full submission is complete, your paper will undergo a series of checks in preparation for peer review. After your manuscript has passed the checks it will be sent out for review. To provide the metadata for your submission, please Login to Editorial Manager (https://www.editorialmanager.com/pbiology) within two working days, i.e. by Mar 21 2023 11:59PM.

Kind regards,

Luke

Lucas Smith, Ph.D.

Associate Editor

PLOS Biology

lsmith@plos.org

---

## [Decision Letter · Decision Letter 1]

23 May 2023

Dear Eunjoon,

Thank you for your patience while your manuscript "Ependymal ciliary hyperfunction in Katnal2-mutant mice with hydrocephalus and ASD-related social, synaptic, and transcriptomic deficits" was peer-reviewed at PLOS Biology, and apologies again for the protracted review process. Your study has now been evaluated by the PLOS Biology editors, an Academic Editor with relevant expertise, and by several independent reviewers. In light of the reviews, which you will find at the end of this email, we would like to invite you to revise the work to thoroughly address the reviewers' reports.

As you will see below, the reviewers find the study interesting and generally well done. However they have also highlighted that further work is necessary to flesh out and support the conclusions, and we think these concerns would need to be carefully addressed, including by providing further controls in response to Reviewer 1. As a note, Reviewer 3 has suggested that the behavioral data be removed. We would be OK with that, but we will ultimately leave it up to you to decide whether or not to include it. 

Given the extent of revision needed, we cannot make a decision about publication until we have seen the revised manuscript and your response to the reviewers' comments. Your revised manuscript is likely to be sent for further evaluation by all or a subset of the reviewers.

**IMPORTANT - SUBMITTING YOUR REVISION**

*Re-submission Checklist*

*Published Peer Review*

*PLOS Data Policy*

*Blot and Gel Data Policy*

Sincerely,

Luke

Lucas Smith, Ph.D.

Associate Editor

PLOS Biology

lsmith@plos.org

REVIEWS:

Reviewer #1: Katanin, a microtubule-severing ATPase, is a protein complex composed of the regulatory subunit Katanin P80 (KATNB1), the catalytic subunit Katanin P60 (KATNA1), and two additional Katanin P60-like proteins (KATNAL1 and KATNAL2). A recent study of Katnal1 found that homozygous missense mutations in Kantal1 caused ventriculomegaly due to abnormally long ependymal cilia with impaired motility (Banks et al., 2018). In this study, the authors claim that Katnal2 KO also results in increased length of ependymal cilia; however, the motility of ependymal cilia is enhanced, raising hydraulic pressure within the ventricular system to cause ventriculomegaly. Although ventriculomegaly and hydrocephalus have been previously described in mutation of other Katanin family members, such as Katnal1 and Katnb1, a strength of this manuscript is that Katnal2 mutation has not yet been linked with ventriculomegaly. However, major weaknesses of this manuscript are that the data and analyses do not support their hypothesis and claims on the mechanism of ventriculomegaly. Due to this, my recommended course of action is rejection of this manuscript. 

First, experiments assessing ependymal cilia function (Fig. 4D-4G) lack control conditions (e.g. motile cilia paralyzing toxins). These experiments are required to determine whether the measured fluorescent bead speeds reflect the motility of ependymal cilia. Further, ependymal cilia function at the anterior ventral portion of the lateral ventricle wall, a critical bottleneck of ependymal cilia flow within the lateral ventricles, was not assessed, which would be required for the complete evaluation of ependymal cilia function. Additionally, the authors do not provide sufficient data to support the claim that increased ependymal cilia motility causes an increase in hydraulic pressure. Specifically, the authors report that increased CSF flow rates in the cerebral aqueduct of anesthetized Katnal2 mutants causes an increase in hydraulic pressure (Fig 4J); however, the measurement of intraventricular pressure would be required to provide support for this claim. The authors show that ventriculomegaly in Katnal2 KO mice is rescued with the delivery of AAV-mediated overexpression of Katnal2 at timepoints before the emergence of hydrocephalus (Fig. 7C); however, analyses of ependymal cilia morphology and function were not performed to validate their hypothesis. Strikingly, the authors show robust endogenous Katnal2 expression within the choroid plexus (Fig. 3C) and remark that this expression may indicate that Katnal2 may regulate the production of CSF; however, no experiments were performed to evaluate a pathological role of the choroid plexus in the etiology of Katnal2-associated ventriculomegaly. This is of fundamental importance because changes in the rate of CSF circulation by ependymal cilia may be secondary to increased production of CSF, not a primary deficit underlying ventriculomegaly. To this point, all of the experiments evaluating ependymal cilia function were performed at timepoints after Katnal2 KO mice have documented ventriculomegaly, and no experiments evaluating ependymal cilia function were done at earlier timepoints when Katnal2 KO mice lack ventriculomegaly (S4A Fig).

Reviewer #2, Froylan Calderon de Anda (Note, reviewer 2 has signed this review): This is an interesting and well-executed study that shows that deficiency of Katnal2 is instrumental in progressive ventricular enlargement. Elegantly, authors show that lack of Katnal2 increased the length and beating frequency of motile cilia on ependymal cells lining ventricles; thus, affecting the ventricular hydraulic pressure. Moreover, it is shown that progressive increment of ventricles leads to decrease cerebral blood volumes in various brain regions. Authors suggest that these volumetric changes might affect neuronal functioning and, accordingly, they show progressive synaptic deficits in the hippocampus that are paired with transcriptomic changes. Finally, it is revealed that mice lacking Katnal2 show selective abnormalities in social communication such as ultrasonic vocalizations in the adult and early in the development (pup stage) that are partially overcome upon the ventricular introduction of Katnal2 (postnatal day 8-12).

The paper is well-written, although it could be organized differently with the behavioral analysis at the end together with the rescue approach (my personal taste). The authors use an interesting approach to test their hypothesis, supplemented with a blend of imaging techniques, and the data is nicely presented. An interesting new link between cilia functioning and ASD is identified; however, the mechanism by which cilia abnormalities and brain volumetric changes exert their effect on synapse physiology is unclear.

Having said this, a number of issues make the interpretation of the results difficult and I consider that it would be beneficial if the authors revise their manuscript.

1. I am wondering why the pups show abnormalities in social communication given that the cellular and molecular parameters tested revealed progressive development of anomalies (not present in the pups). 

2. Even though the cellular architecture of the cortex seems to be not affected (cortex layering), suggesting no issues with neurogenesis and neuronal migration, it could be helpful to analyze how neuronal morphology is affected when they lack this katanin-like protein. We know that microtubule dynamics are key during neuronal differentiation. Moreover, neurons have cilia, and little is known about this organelle in neurons. Maybe culturing neurons (or analysis in situ from embryonic tissue) from mice lacking Katnal2 may teach us something that could explain the phenotypes observed in the pups.

Reviewer #3: Eunjoon Kim and colleagues present a compelling study that highlights the contributions of the Katnal2 gene in molecular, cellular, and behavioral phenotypes related to Autism Spectrum Disorder (ASD). The authors utilized Katnal2 knockout mice to demonstrate that absence of Katnal2 results in autistic-like behaviors, particularly social communication deficits. These behavioral changes were accompanied by age-dependent ventricular enlargements, elongated ependymal cilia, and increased ciliary beating frequency, which were correlated with the expression of Katnal2 in the ependymal cilia and other brain regions. The study also observed impairments in synaptic functions and transcriptomic signature. Interestingly, re-integration of Katnal2 early postnatally prevented ventricular and behavioral deficits in the adult mice, suggesting a possible link between Katnal2 (dys)function and the observed phenotypes. 

The creation of the Katnal2 KO mouse model and the involvement of this gene in the function of the cilia and ventricular dysfunction is a valuable contribution to the scientific community. While the data presented in the study warrant publication in PLOS Biology, I suggest removing the behavioural part that does not help to clarify the role of Katnal2 in the functioning of the cilia and brain development. For example at a juvenile age it seems that the social deficits (USV) observed are independent from the ventricules malformation since they are OK. I am also wondering why the number of mice is quite low for some behaviours and very high for others.

---

## [Decision Letter · Decision Letter 2]

15 Dec 2023

Dear Eunjoon, 

Thank you for your patience while we considered your revised manuscript "Ependymal ciliary hyperfunction in Katnal2-mutant mice with hydrocephalus and ASD-related social, synaptic, and transcriptomic changes" for publication as a Research Article at PLOS Biology. Your revised study has been evaluated by the PLOS Biology editors, the Academic Editor and the original reviewers.

The reviewer comments are appended below and you will see that reviewers 2 and 3 are largely satisfied by the revision, and they have suggested we accept the study. However, reviewer 1 has a number of lingering concerns. We would therefore like to invite you to revise the work further, to address the last of reviewer 1's comments.

Please note that, we would editorially overrule reviewer 1's concerns about a recent and related preprint, and its impact on the current work. We do not think that a preprint impacts the novelty of your work (and we also have a 'scooping protection policy') - and we think that is is OK that your findings differ from other reports on this topic. Further work, beyond the scope of this study, will be needed to interrogate these differences. After discussion with the Academic Editor, we think that, in large part, Reviewer 1's remaining concerns can be addressed with textual changes. However, we agree with Reviewer 1 that more experimentfal work is needed to assess changes in CSF production directly. The reviewer has suggested some examples of additional work that might address this point. 

Given the extent of revision needed, we cannot make a decision about publication until we have seen the revised manuscript and your response to the reviewers' comments. Your revised manuscript may be sent for further evaluation by the reviewers.

While the scope of the required revision is fairly narrow, as we anticipate additional experiments will be needed to address these last issues, we have provided you with a 3 month deadline for your revision. Please email us (plosbiology@plos.org) if you have any questions or concerns, or would like to request an extension. 

**IMPORTANT - SUBMITTING YOUR REVISION**

*Re-submission Checklist*

*Published Peer Review*

*PLOS Data Policy*

*Blot and Gel Data Policy*

Sincerely,

Luke

Lucas Smith, Ph.D.

Senior Editor

PLOS Biology

lsmith@plos.org

REVIEWS:

Reviewer #1: A recent pre-print has been published by Qiu et al., on BioRxiv (doi: https://doi.org/10.1101/2023.07.03.547302), delineating the mechanism of ventricular enlargement in a Katnal2 mutant mouse model with conditional deletion of Katnal2 specifically within epithelial cells of the ependyma and choroid plexus of neonatal mice. Qui et al. report that Katnal2 mutant mice have decreased ependymal cilia beating and reduced CSF flow, consistent with previously published findings in Katnal1 mutant mice with ventriculomegaly. These data contradict data presented here by Kang et al. showing increased ependymal cilia beating frequency and increased CSF flow rates despite normal ICP. Major concerns remain about the mechanism of ventriculomegaly, and the recently published pre-print decreases novelty of this finding in Katnal2 mutant mice. 

Due to this, my recommended course of action is rejection of this manuscript.

First, experiments assessing ependymal cilia function (Fig. 4D-4G) lack control conditions (e.g. motile cilia paralyzing toxins). These experiments are required to determine whether the measured fluorescent bead speeds reflect the motility of ependymal cilia. 

We appreciate the comment and agree that this is an essential negative control. In response, we performed the bead motility experiment after 10-min incubation of the brain slices with pneumolysin, a pneumococcal toxin known to inhibit motile cilia in a dose-dependent manner [1, 2]. This treatment significantly reduced the motility of the beads (Fig 3H), supporting the validity of the bead motility assay. Further, ependymal cilia function at the anterior ventral portion of the lateral ventricle wall, a critical bottleneck of ependymal cilia flow within the lateral ventricles, was not assessed, which would be required for the complete evaluation of ependymal cilia function. � In response, we analyzed the bead motility in the anterior ventral portion of the lateral ventricular wall. Similar to the results from the AD/PM areas, the bead motility in the anterior ventral portion was also increased in Katnal2-KO mice (Fig 3E).

-The inclusion of a toxin washout experiment to show restoration of ependymal cilia function would have been the proper experiment to validate the bead motility assay.

Additionally, the authors do not provide sufficient data to support the claim that increased ependymal cilia motility causes an increase in hydraulic pressure. Specifically, the authors report that increased CSF flow rates in the cerebral aqueduct of anesthetized Katnal2 mutants causes an increase in hydraulic pressure (Fig 4J); however, the measurement of intraventricular pressure would be required to provide support for this claim. 

In response, we measured the intracranial pressure (ICP) in Katnal2-KO mice (3 months), using the pressure sensor previously described [3]. The results indicated comparable ICPs in WT and Katnal2-KO mice (Fig 3L). We think that the increased ICP in Katnal-2 KO mice might have been counteracted by the increased ventricular volumes. In line with this, about half of the patients with idiopathic ventriculomegaly show normal ICPs. We commented on this in the revised Results and Discussion.

-The authors provide data that Katnal2-KO mice have comparable ICPs to WT mice. It is highly unlikely that increased ependymal cilia motility causes increased hydraulic pressure without also raising intraventricular pressure. This mechanism has never been described in mice or humans with ventriculomegaly and is inconsistent with previous findings in Katnal1-KO mice and the recent Katnal2-KO pre-print manuscript. 

The authors show that ventriculomegaly in Katnal2 KO mice is rescued with the delivery of AAV-mediated overexpression of Katnal2 at timepoints before the emergence of hydrocephalus (Fig. 7C); however, analyses of ependymal cilia morphology and function were not performed to validate their hypothesis.

In response, we tested if the increased length of ependymal cilia in Katnal2-KO mice could be restored by Katnal2 re-expression and found that the normalization of the ciliary length in Katnal2-KO mice with Katnal2 re-expression to levels comparable to those in wild-type mice (Fig 6C).

-The authors provide data that Katnal2 overexpression restores ciliary length; however, the functional analysis of these cilia with normalized length was not assessed.

Strikingly, the authors show robust endogenous Katnal2 expression within the choroid plexus (Fig. 3C) and remark that this expression may indicate that Katnal2 may regulate the production of CSF; however, no experiments were performed to evaluate a pathological role of the choroid plexus in the etiology of Katnal2- associated ventriculomegaly. This is of fundamental importance because changes in the rate of CSF circulation by ependymal cilia may be secondary to increased production of CSF, not a primary deficit underlying ventriculomegaly.

In response, we re-analyzed blood perfusion levels in the choroid plexus. CBVweighted signals around the choroid plexus were unaffected, (Fig EV7B,C), suggesting that CSF production, relying on the blood flow to the choroid plexus [4, 5], was not changed.

-The authors examine blood perfusion levels in the choroid plexus as a proxy for CSF production; however, this is an inadequate assessment of CSF production. It has previously been shown that the cilia of choroid plexus epithelial cells regulate CSF production. Despite robust Katnal2 expression within the choroid plexus, the authors examined cilia length of ependymal epithelial cells without also examining cilia length of epithelial cells of the choroid plexus. At minimum, the authors should have examined the expression of ion co-transporters (Na+-K+-ATPase, AE2) and water channels (AQP1) in Katnal2-KO mice. 

Reviewer #2, Froylan Calderon de Anda (Note - reviewer 2 has signed this review): The authors properly answered my question; thus, I consider is acceptable for publication

Reviewer #3: I appreciate the authors additional experiments added to the manuscript. 

However, I maintain my viewpoint that keeping the data on social, even to the supplementary section, diminishes the paper's strength by diverting attention from its core focus. 

I leave it to the discretion of the authors and the editor to decide whether they wish to retain it. 

Nevertheless, I support the publication of the manuscript, the scientific community will benefit of a new mouse model that could enhance our understanding of one of the various functions of cilia.

---

## [Editor Report · Decision Letter 3]

18 Mar 2024

Dear Eunjoon,

Thank you for your patience while we considered your revised manuscript "Ependymal ciliary hyperfunction in Katnal2-mutant mice with hydrocephalus and ASD-related social, synaptic, and transcriptomic changes" for publication as a Research Article at PLOS Biology and apologies for the time it has taken to send you a decision. These last few weeks have been very busy on my end which has caused the delay! This revised version of your manuscript has now been evaluated by the PLOS Biology editors, and by the Academic Editor, who is satisfied by your response to the last reviewer requests.

Based on our Academic Editor's assessment of your revision, we are likely to accept this manuscript for publication. However, before we can accept your study, we need you to address a number of editorial requests in a revision that we anticipate will not take very long. These are detailed below. 

**EDITORIAL REQUESTS: 

1) TITLE: We would like to suggest some changes to the title which we think will improve its clarity. If you agree, we suggest the title be changed to something like: 

"Loss of Katnal2 leads to ependymal ciliary hyperfunction and autism-related phenotypes in mice"

2) ETHICS STATEMENT: Thank you for providing an ethics statement in your methods section. Please update this to include the specific national or international regulations/guidelines to which your animal care and use protocol adhered. Please note that institutional or accreditation organization guidelines (such as AAALAC) do not meet this requirement.

3) DATA: Thank you for providing the RNA-seq data as a deposition to GEO. I noticed this is currently private. As a heads up, this dataset will need to be made public upon publication. 

4) DATA: You may be aware of the PLOS Data Policy, which requires that all data be made available without restriction: http://journals.plos.org/plosbiology/s/data-availability. For more information, please also see this editorial: http://dx.doi.org/10.1371/journal.pbio.1001797

While the GEO dataset meets part of this requirement, we also need the underlying data for the other experiments presented here. Note that we do not require all raw data. Rather, we ask that all individual quantitative observations that underlie the data summarized in the figures and results of your paper be made available in one of the following forms:

a. Supplementary files (e.g., excel). Please ensure that all data files are uploaded as 'Supporting Information' and are invariably referred to (in the manuscript, figure legends, and the Description field when uploading your files) using the following format verbatim: S1 Data, S2 Data, etc. Multiple panels of a single or even several figures can be included as multiple sheets in one excel file that is saved using exactly the following convention: S1_Data.xlsx (using an underscore).

b. Deposition in a publicly available repository. Please also provide the accession code or a reviewer link so that we may view your data before publication. 

>>Regardless of the method selected, please ensure that you provide the individual numerical values that underlie the summary data displayed in the following figure panels as they are essential for readers to assess your analysis and to reproduce it:

Fig 1A,C,D; Fig 3B,E,G-H,I,L; Fig 4C,D,E-N; Fig 6C,D,E;

Fig S1E,H-I; FigS2; Fig S3; Fig S4; Fig S7A-C,E;

>>Please also ensure that figure legends in your manuscript include information on where the underlying data can be found, and ensure your supplemental data file/s has a legend.

>>Please ensure that your Data Statement in the submission system accurately describes where your data can be found.

5) CODE: I see you have generated some code, provided on Github. Can you please update Github to add a readme file, providing some relevant details of what this code is, and how it relates to the study? Please also update your 'Data Availability Statement' in our online system to reference all newly generated code. 

6) METHODS: I see that some of the methods are currently presented in the supplement. Can you please move these details into the main text?

7) OTHER: As a last note, the figures in your 'supplementary information' file seem up to date, but I noticed you have 2 Supplemental Figure 7's in the file inventory. As the figures in the file inventory will ultimately be pulled into the final publication, please do go through all the figures provided and make sure they are the most up-to-date version. 

We expect to receive your revised manuscript within two weeks. 

*Published Peer Review History*

*Press*

Sincerely,

Luke 

Lucas Smith, Ph.D.

Senior Editor

lsmith@plos.org

PLOS Biology

---

## [Editor Report · Decision Letter 4]

21 Mar 2024

Dear Eunjoon,

Thank you for the submission of your revised Research Article "Loss of Katnal2 leads to ependymal ciliary hyperfunction and autism-related phenotypes in mice" for publication in PLOS Biology and thank you for addressing our last editorial requests. On behalf of my colleagues and the Academic Editor, Richard Daneman, I am pleased to say that we can in principle accept your manuscript for publication, provided you address any remaining formatting and reporting issues. These will be detailed in an email you should receive within 2-3 business days from our colleagues in the journal operations team; no action is required from you until then. Please note that we will not be able to formally accept your manuscript and schedule it for publication until you have completed any requested changes.

PRESS

Sincerely, 

Lucas Smith, Ph.D.

Senior Editor

PLOS Biology

lsmith@plos.org